# Bayesian Inference and Data Analysis of the Unit–Power Burr X Distribution

**Aisha Fayomi** [1] (ID)**, Amal S. Hassan** [2],*(ID)**, Hanan Baaqeel** [1] **and Ehab M. Almetwally** [3],*

[1] Department of Statistics, Faculty of Science, King Abdulaziz University, Jeddah 21589, Saudi Arabia
[2] Faculty of Graduate Studies for Statistical Research (FGSSR), Cairo University, Giza 12613, Egypt
[3] Faculty of Business Administration, Delta University for Science and Technology, Gamasa 11152, Egypt
* Correspondence: amal52_soliman@cu.edu.eg (A.S.H.); ehab.metwaly@deltauniv.edu.eg (E.M.A.)

**Abstract:** The unit–power Burr X distribution (UPBXD), a bounded version of the power Burr X distribution, is presented. The UPBXD is produced through the inverse exponential transformation of the power Burr X distribution, which is also beneficial for modelling data on the unit interval. Comprehensive analysis of its key characteristics is performed, including shape analysis of the primary functions, analytical expression for moments, quantile function, incomplete moments, stochastic ordering, and stress–strength reliability. Rényi, Havrda and Charvat, and d-generalized entropies, which are measures of uncertainty, are also obtained. The model's parameters are estimated using a Bayesian estimation approach via symmetric and asymmetric loss functions. The Bayesian credible intervals are constructed based on the marginal posterior distribution. Monte Carlo simulation research is intended to test the accuracy of various estimators based on certain measures, in accordance with the complex forms of Bayesian estimators. Finally, we show that the new distribution is more appropriate than certain other competing models, according to their application for COVID-19 in Saudi Arabia and the United Kingdom.

**Keywords:** power Burr X distribution; entropy; Bayesian estimation; Metropolis–Hastings; COVID-19 data





## 1. Introduction

Utilizing differential equations, Burr [1] introduced twelve distributions. In the literature, Burr type XII distributions and single-parameter Burr type X, have drawn much interest. Surles and Padgett [2] have proposed the two-parameter Burr X distribution (BXD), often known as the generalized Rayleigh distribution. For data modelling, the BXD can be used as an alternative to the Weibull and Rayleigh distributions. However, the model has a considerable impact on the prediction of failure rates and has generated a lot of interest in modelling across a wide range of disciplines, including hydrology, medicine and reliability analysis. The cumulative distribution function (CDF) of the BXD is given by:

$$G(x) = 1 - \exp{-(a\,x^b)^2}; \quad x > 0, \tag{1}$$

where, $b > 0$, and $a > 0$ are the shape and scale parameters, respectively. The probability density function (PDF) associated with (1) is given by:

$$g(x) = 2a^2bx^{2b-1} \exp{-(a\,x^b)^2}; \quad x > 0. \tag{2}$$

According to Raqab and Kundu [3], the shape parameter ($b$) determines whether the hazard rate function (HF) of the BXD is a bathtub or an increasing function. The HF is bathtub for $b \le 1$, and is an increasing function for $b > 1$. In the literature, numerous studies have been undertaken in recent years to create modified or generalized forms of

the BXD in order to increase the viability of BXDs, see, for example, [4–9]. Our focus is on the recently established power BXD (PBXD) by Usman and Ilyas [10], with an additional shape parameter that depends on the transformation $Y = 1/X^{1/\delta}$, $\delta > 0$. The CDF and PDF of the PBXD are, respectively, given by:

$$G(y) = \left[1 - \exp{-(a\,y^b)^2}\right]^{\delta}; \ y > 0, \tag{3}$$

$$g(y) = 2\delta a^2 b y^{2b-1} e^{-(a\,y^b)^2} \left[1 - \exp{-(a\,y^b)^2}\right]^{\delta-1}; \ y > 0. \tag{4}$$

For $\delta = 1$, the CDF (3) reduces to BXD. Usman and Ilyas [10] mentioned that, subject to certain restrictions, their model can handle both symmetrical and heavy-tailed skewed data sets.

A significant challenge in data modelling is the selection of an adequate lifetime probability. However, over time, a variety of probability models have been widely proposed for the analysis of data sets in a variety of fields, including the medical sciences, actuarial sciences, engineering, finance and insurance, demography, biological sciences, and economics. In many practical scenarios, we are required to deal with the uncertainty of bounded situations. We commonly encounter variables that fall within the range of (0, 1), such as the percentage of a particular trademark, the results of some capacity tests, different lists, and rates. In order to model these variables effectively, continuous unit distributions, or probability distributions with support for (0, 1), are crucial. Due to this, some authors have recently concentrated on the creation of distributions that are specified on the bounded interval using any one of the parent distribution modification strategies. Among distributions that are specified in the (0, 1) interval, the beta distribution is obviously the most well-known. The beta distribution is helpful for simulating data on the unit interval, but different distributions have also been proposed and researched over time. The Topp–Leone distribution (see [11]) and the Kumaraswamy distribution (see [12]) can all be used as examples by the reader. The idea of offering distributions defined by the unit interval corresponding to any continuous distribution, however, has recently attracted the interest of statisticians. The following are a few of the most practical unit–interval distributions: the log–Lindley (Gómez-Déniz et al. [13]), unit–Birnbaum–Saunders (Mazucheli et al. [14]), unit–inverse Gaussian (Ghitany et al. [15]), unit–Lindley (Mazucheli et al. [16]), unit–BurrIII (Modi and Gill [17]), unit–Weibull (Mazucheli et al. [18]), unit–Burr XII (Korkmaz and Chesneau [19]), unit–odd Fréchet power function (Haq et al. [20]), unit–Teissier (Krishna et al. [21]), unit–exponentiated exponential (Jha et al. [22]) and unit–exponentiated half-logistic (Hassan et al. [23]) among others.

In this study, we propose a new unit probability distribution, based on the PBXD, that has three parameters. A new unit-PBXD (UPBXD) is provided based on the transformation $W = e^{-Y}$, where $Y$ represents the PBXD. The UPBXD has the following desirable characteristics:

- The UPBXD is a flexible model and can be used to describe a variety of datasets with a range between zero and one.
- The new density function of the UBBXD takes several shapes, including unimodal, reversed J-shaped, U-shaped, left-skewed, and symmetric (see Section 2).
- The HF shapes of the UPBXD can be increasing, J-shaped, or bathtub (U-HF) (see Section 2).
- We derive some of the most important statistical characteristics of the UPBXD, such as the analytical expression for moments, the quantile function, incomplete moments, stochastic ordering, some uncertainty measures, and stress–strength reliability.
- The parameter estimators of the UPBXD are explored using a Bayesian technique. The Bayesian credible intervals are also created.
- To examine the effectiveness of estimators based on accuracy criteria, an exclusive simulation study was conducted.
- Application to COVID-19 datasets from Saudi Arabia and the United Kingdom are used to show the superiority of the proposed model over other well-known models.

An outline of the paper's structure is provided. Section 2 provides a definition of the suggested distribution. The distributional characteristics of the UPBXD are covered in Section 3. The maximum likelihood (ML) and Bayesian estimators utilizing various loss functions are covered in Section 4. The effectiveness of the suggested point and interval estimators is assessed using a Monte Carlo simulation in Section 5. Section 6 shows that the UPBXD outperforms the other unit distributions when employed with COVID-19 data. The paper conclusion is completed in Section 7.

## 2. Unit Power Burr X Distribution

In this section, we present the UPBXD, which results from the transformation of the type $W = e^{-Y}$, where $Y$ is the PBXD and is a new bounded distribution with support on $(0, 1)$. Thus, the following is how the CDF of the PBXD can be obtained:

$$F(w) = P(W \leq w) = P(e^{-Y} \leq w) = P(-Y \leq \ln(w)) = 1 - P(Y \leq -\ln(w)) = 1 - F_Y(-\ln(w)),$$

which gives

$$F(w) = 1 - \left\{ 1 - \exp\left[ -\left( a\left(-\ln w\right)^b \right)^2 \right] \right\}^\delta; \quad 0 < w < 1, \, a, b, \delta > 0. \tag{5}$$

Based on (5), we have $F(w) = 0$, for $w \leq 0$, and $F(w) = 1$, for $w \leq 1$. The PDF of the UPBXD related to (5) can be acquired as follows:

$$f(w) = 2a^2 b\delta w^{-1}(-\ln w)^{2b-1} e^{-(a\left(-\ln w\right)^b)^2} \left\{ 1 - \exp\left[ -\left( a\left(-\ln w\right)^b \right)^2 \right] \right\}^{\delta-1}; \quad 0 < w < 1. \tag{6}$$

A random variable with PDF (6) is represented by UPBXD $(a, b, \delta)$. For $b = 1$, the PDF (6) gives UBXD as a new sub-model. The following is the HF of the UPBXD:

$$h(w) = 2a^2 b\delta w^{-1}(-\ln w)^{2b-1} e^{-(a\left(-\ln w\right)^b)^2} \left\{ 1 - \exp\left[ -\left( a\left(-\ln w\right)^b \right)^2 \right] \right\}^{-1}. \tag{7}$$

The related plots for various selections of the parameters $a, b$, and $\delta$ are shown in Figures 1 and 2 to provide a general overview of the shapes of the PDF (2) and HF (7).

In Figure 1, the PDF graphs for various parameter combinations display a variety of shapes, such as ($a = 2$, $b = 2$) symmetric normal, ($a = 0.5$, $b = 0.3$) U-shaped, ($a = 0.5$, $b = 2$) right-skewed, ($a = 2$, $b = 0.3$) J-shaped, and ($a = 2$, $b = 2$) normal tapered. In Figure 2, the UPBXD's HF shapes in ($a = 0.5$, $b = 2$), ($a = 2$, $b = 0.3$), and ($a = 2$, $b = 2$) have increasing and J shapes, while ($a = 0.5$, $b = 0.3$) has a bathtub shape.

The parameter $\delta$ is responsible for the bathtub shapes given that the other two parameters ($a$ and $b$) are less than one. The $\delta$ parameter is responsible for the J shapes where $a > 1$ and $b < 1$.

By inverting (5), we can get the quantile function (QF) of the UPBXD, which looks like this:

$$w_q = \exp\left( -\left\{ \frac{1}{a}\left[ -\ln\left\{ 1 - (1-q)^{1/\delta} \right\}^{\frac{1}{2}} \right] \right\}^{1/b} \right), \quad 0 < q < 1, \tag{8}$$

where $q$ is the uniform random variables. The first, median, and third quantiles are produced by setting $q = 0.25$, $0.5$, and $0.75$ in (8). It is simple to simulate the random variable of the UPBXD from (8).

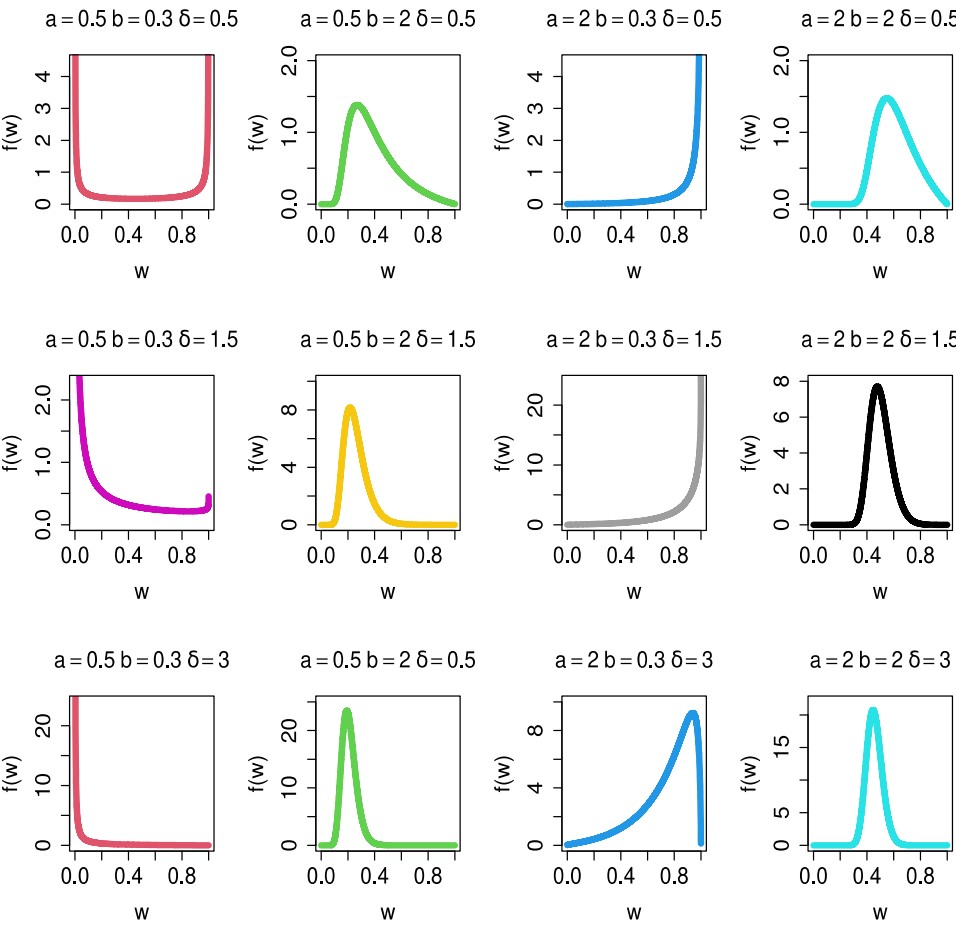

**Figure 1.** Plots of various PDF shapes of the UPBXD for different parameter values.

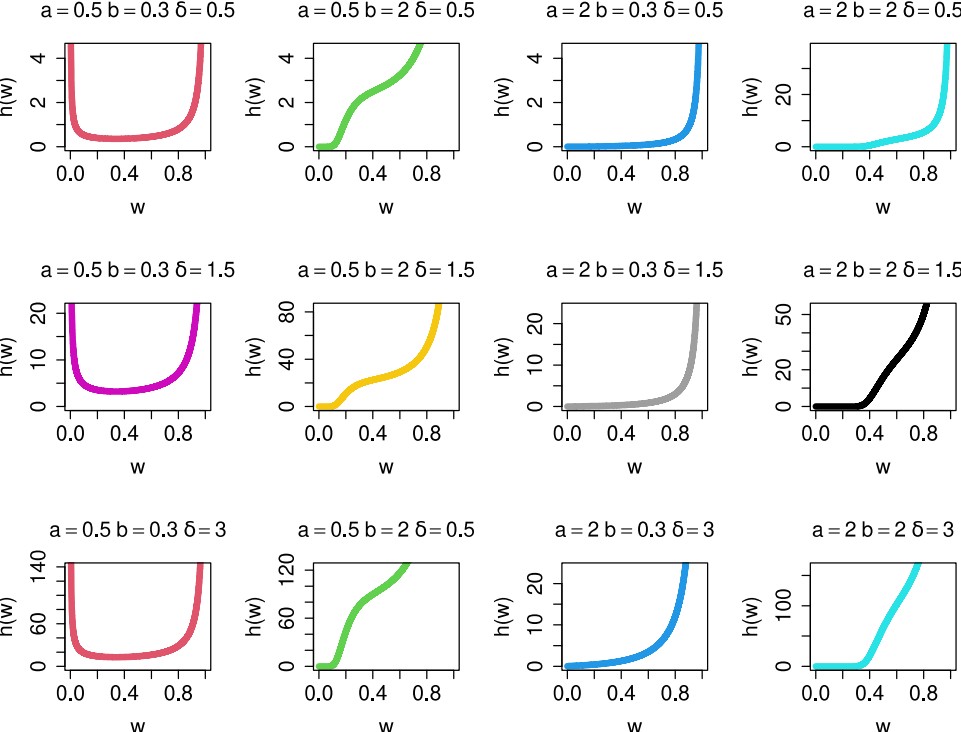

**Figure 2.** Plots of various HF shapes of the UPBXD for different parameter values.

## 3. The UPBXD's Properties

In this section, we examine aspects of the UPBXD's structural characteristics, such as some moment's measures, information measures, stochastic ordering (SO), and stress–strength (SS) reliability.

### 3.1. Some Moments Measures

The $m$th moment for $W \sim$ UPBXD $(a, b, \delta)$, is determined as follows:

$$\mu'_m = 2a^2 b \delta \int_0^1 w^{m-1} (-\ln w)^{2b-1} e^{-(a(-\ln w)^b)^2} \left[ 1 - \exp\left[ -\left( a(-\ln w)^b \right)^2 \right] \right]^{\delta-1} dw. \quad (9)$$

Using the binomial expansion in (9) provides

$$\mu'_m = 2a^2 b \delta \sum_{j=0}^{\infty} (-1)^j \binom{\delta-1}{j} \int_0^1 w^{m-1} (-\ln w)^{2b-1} e^{-(j+1)(a(-\ln w)^b)^2} dw.$$

Let $y = \left( a(-\ln w)^b \right)^2$, then the $m$th moment of $W$, is given by

$$\mu'_m = \delta \sum_{j=0}^{\infty} (-1)^j \binom{\delta-1}{j} \int_0^{\infty} e^{-m(a)^{\left(\frac{-1}{b}\right)} y^{1/2b}} e^{-(j+1)y} dy.$$

Use the exponential expansion then $\mu'_m$, obtains the following form:

$$\begin{aligned}
\mu'_m &= \sum_{j,k=0}^{\infty} \frac{(-1)^{j+k} m^k \delta}{k!} a^{\frac{-k}{b}} \binom{\delta-1}{j} \int_0^{\infty} y^{\frac{k}{2b}} e^{-(j+1)y} dy \\
&= \sum_{j,k=0}^{\infty} \frac{(-1)^{j+k} \delta m^k}{k!(j+1)^{(k/2b)+1}} a^{\frac{-k}{b}} \binom{\delta-1}{j} \Gamma\left( \frac{k}{2b} + 1 \right),
\end{aligned}$$

where, $\Gamma(.)$ is a gamma function. Furthermore, the $m$th central moment of $W$, is defined by

$$\mu_m = E(W - \mu'_1)^m = \sum_{i=0}^{m} (-1)^i \binom{m}{i} (\mu'_1)^i \mu'_{m-i}.$$

Some moments measures including, first four moments, variance ($\sigma^2$), coefficient of skewness ($\alpha_3$) and coefficient of kurtosis ($\alpha_4$) for the UPBXD are calculated for specific parameter values. Table 1 provides these measures considering parameter values as: (i) $(a = 1.5, b = 1.3, \delta = 1.4)$, (ii) $(a = 0.5, b = 0.4, \delta = 0.4)$, (iii) $(a = 4, b = 2, \delta = 0.7)$, (iv) $(a = 0.7, b = 0.4, \delta = 2)$, (v) $(a = 1.5, b = 0.5, \delta = 0.5)$, (vi) $(a = 5, b = 1.6, \delta = 0.4)$, and (vii) $(a = 1.5, b = 1.5, \delta = 3)$.

**Table 1.** Several UPBXD moment values.

| $\mu'_m$ | (i) | (ii) | (iii) | (iv) | (v) | (vi) | (vii) |
|---|---|---|---|---|---|---|---|
| $\mu'_1$ | 0.499 | 0.396 | 0.67 | 0.802 | 0.803 | 0.477 | 0.416 |
| $\mu'_2$ | 0.264 | 0.231 | 0.458 | 0.653 | 0.688 | 0.239 | 0.18 |
| $\mu'_3$ | 0.148 | 0.160 | 0.32 | 0.54 | 0.612 | 0.125 | 0.08 |
| $\mu'_4$ | 0.087 | 0.122 | 0.228 | 0.454 | 0.556 | 0.069 | 0.037 |
| $\sigma^2$ | 0.015 | 0.074 | 0.009 | 0.011 | 0.044 | 0.011 | 0.0062 |
| $\alpha_3$ | 0.377 | 0.510 | 0.38 | −0.135 | −1.237 | 0.465 | 0.375 |
| $\alpha_4$ | 2.856 | 2.132 | 2.75 | 2.248 | 3.799 | 3.070 | 3.135 |

Table 1 displays that the UPBXD is right- and left- skewed in accordance with the values of $\alpha_3$. Additionally, the distribution is leptokurtic and platykurtic according to the values of $\alpha_4$. Figure 3 shows the 3-dimensional plots for coefficient of skewness and kurtosis for UPBXD with different values of parameters. Looking at Figure 3, we can see that the coefficient of skewness and kurtosis increases when *b* and $\delta$ increases, while *a* increases then the coefficient of skewness decreases and coefficient of kurtosis increases.

Furthermore, the *m*th lower incomplete moment, say $v_m(x)$, of the UPBXD is given by:

$$v_m(x) = 2a^2b\delta \int_0^x w^{m-1}(-\ln w)^{2b-1} e^{-(a(-\ln w)^b)^2} \left[1 - \exp\left[-\left(a(-\ln w)^b\right)^2\right]\right]^{\delta-1} dw.$$

Let $z = \left(a(-\ln w)^b\right)^2$, and using the binomial expansion, then the *m*th incomplete moment of *W* is

$$v_m(x) = \delta \sum_{j=0}^{\infty} (-1)^j \binom{\delta-1}{j} \int_{(a(-\ln x)^b)^2}^{\infty} e^{-m(a)^{(\frac{-1}{b})} y^{1/2b}} e^{-(j+1)y} dy.$$

Using exponential expansion and after simplification, the *m*th moment is as below:

$$v_m(x) = \sum_{j,k=0}^{\infty} \frac{(-1)^{j+k} \delta m^k}{k!(j+1)^{(k/2b)+1}} a^{\frac{-k}{b}} \binom{\delta-1}{j} \gamma\left(\frac{k}{2b} + 1, (j+1)(a(-\ln x)^b)^2\right),$$

where $\gamma(.,x)$ is an upper incomplete gamma function. The Lorenz and Bonferroni curves are well-known applications of the first incomplete moment. In the fields of economics, demographics, insurance, engineering, and medicine, these curves are especially helpful.

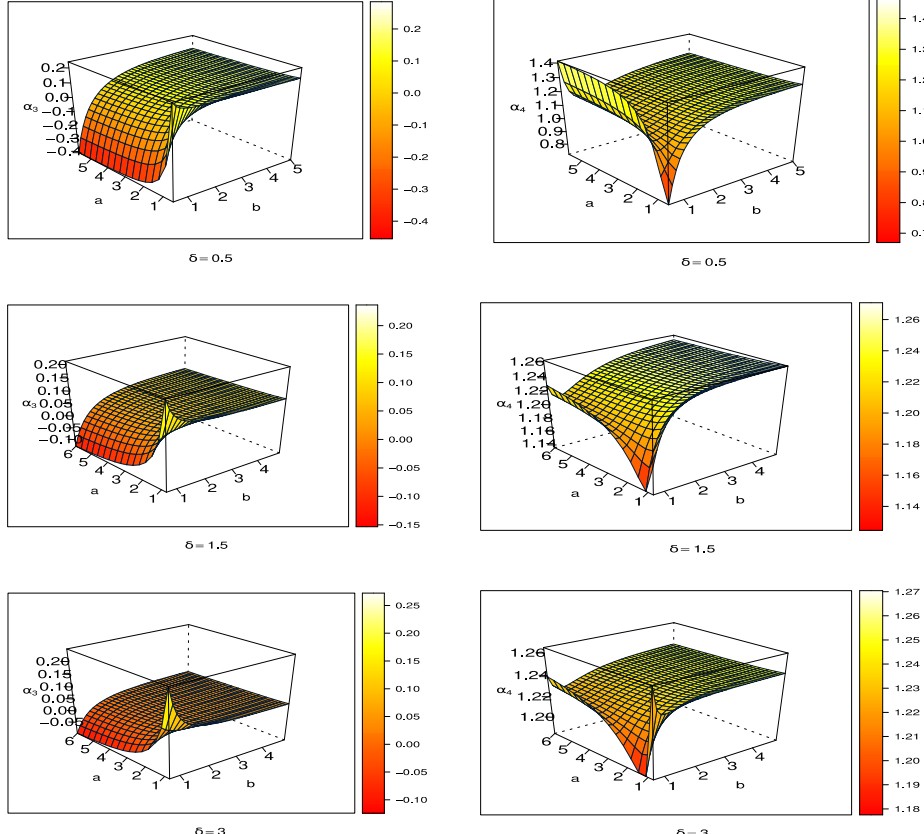

**Figure 3.** Coefficient of skewness and kurtosis for UPBXD.

### 3.2. Information Measures

In this sub-section, we examine the entropies of Rényi, Havrda and Charvat, as well as $d$-generalized entropy as information metrics. These measures collectively provide information about the system's overall amounts of data. The Rényi entropy presented by Rényi [24], is conceptually the quantity of information contained in a random process, it is defined by:

$$\eta(d) = (1-d)^{-1} \log\left( \int_{-\infty}^{\infty} (f(w))^d dw \right), d > 0, d \neq 1. \tag{10}$$

Inserting (6) in (10), and using binomial expansion, then $\eta(d)$ is as follows:

$$\eta(d) = \frac{1}{(1-d)} \log\left( \sum_{i=0}^{\infty} (-1)^i \binom{d(\delta-1)}{i} (2a^2 b\delta)^d \int_0^1 w^{-d}(-\ln w)^{d(2b-1)} e^{-(d+i)(a(-\ln w)^b)^2} dw \right). \tag{11}$$

Let $z = \left( a\left(-\ln w\right)^b \right)^2$, and using exponential expansion in (11), we obtain

$$\eta(d) = \frac{1}{(1-d)} \log\left( \sum_{i,m=0}^{\infty} \Xi_{i,m}(d,\delta,b,a)\Gamma\left( \frac{d(2b-1)+1+m}{2b} \right) \right), \tag{12}$$

where

$$\Xi_{i,m}(d,\delta,b,a) = (-1)^{i+m}\binom{d(\delta-1)}{i} \frac{(2b)^{d-1}\delta^d(d-1)^m}{(m!)(d+i)^{\frac{d(2b-1)+1+m}{2b}}} a^{\frac{d}{b}-\frac{1}{b}-\frac{m}{b}}$$

Reference [25] proposed another uncertainty measure, the Havrda and Charvat. Here we assume H$(d)$, and this is represented mathematically by:

$$\mathrm{H}(d) = \frac{1}{2^{1-d}-1}\left[ \left( \int_{-\infty}^{\infty} (f(w))^d dw \right)^{\frac{1}{d}} - 1 \right], d \neq 1, d > 0.$$

Using the same procedure above, we obtain H$(d)$, as follows:

$$\mathrm{H}(d) = \frac{1}{2^{1-d}-1}\left[ \left( \sum_{i,m=0}^{\infty} \Xi_{i,m}(d,\delta,b,a)\Gamma\left( \frac{d(2b-1)+1+m}{2b} \right) \right)^{\frac{1}{d}} - 1 \right]$$

In reference [26], a further generalized Shannon entropy form known as $d$-generalized entropy was developed. It is represented mathematically as below:

$$\mathrm{K}(d) = (d-1)^{-1}\left[ \int_{-\infty}^{\infty} (f(w))^{2-d} dw - 1 \right], \quad 0 < d < 2, d \neq 1.$$

Using a similar way as above, we obtain the $d$-generalized entropy as follows,

$$\mathrm{K}(d) = \frac{1}{(d-1)}\left( \sum_{i,m=0}^{\infty} \Xi_{i,m}^{**}(d,\delta,b,a)\Gamma\left( \frac{(2-d)(2b-1)+1+m}{2b} \right) \right),$$

where

$$\Xi_{i,m}^{**}(d,\delta,b,a) = (-1)^{i+m}\binom{(2-d)(\delta-1)}{i} \frac{(2b)^{1-d}\delta^{2-d}(d+1)^m}{(m!)(2-d+i)^{\frac{(2-d)(2b-1)+1+m}{2b}}} a^{\frac{1}{b}-\frac{d}{b}-\frac{m}{b}}$$

We use the following sets of parameters to provide entropy numerical values for the measurements under consideration: (i) $(a = 1.5, b = 1.3, \delta = 1.4)$, (ii) $(a = 0.5, b = 0.4, \delta = 0.4)$, (iii) $(a = 4, b = 2, \delta = 0.7)$, (iv) $(a = 0.7, b = 0.4, \delta = 2)$, (v) $(a = 5, b = 1.6, \delta = 0.4)$,

(vi) ($a = 0.7, b = 0.5, \delta = 2$), and (vii) ($a = 1.5, b = 1.5, \delta = 3$). Table 2 provides some numerical values for the provided three entropy measures.

**Table 2.** Numerical values for the UPBXD's entropy measures.

| $d$ | Measures | (i) | (ii) | (iii) | (iv) | (v) | (vi) | (vii) |
|-----|----------|-----|------|-------|------|-----|------|-------|
| 0.5 | $\eta(d)$ | −0.619 | −0.308 | −0.819 | −0.829 | −0.499 | −0.669 | −0.944 |
|     | H($d$) | −1.115 | −0.64 | −1.35 | −1.36 | −0.948 | −1.177 | −1.475 |
|     | K($d$) | −1.101 | 0.792 | −1.354 | −1.201 | −2.953 | −1.179 | −1.688 |
| 1.5 | $\eta(d)$ | −0.877 | 1.01 | −1.034 | −0.94 | −1.814 | −0.927 | −1.224 |
|     | H($d$) | −1.16 | 0.976 | −1.405 | −1.257 | −2.835 | −1.236 | −1.72 |
|     | K($d$) | −0.533 | −0.286 | −0.672 | −0.678 | −0.442 | −0.568 | −0.753 |

### 3.3. Stochastic Ordering

The statistical literature places a great emphasis on the ordering of distributions, especially among lifetime distributions. A significant part of the ranking of various lifetime distributions is found in Johnson et al. [27]. Here, we take into account four distinct SO for two independent UPBX random variables with a restricted parameter space: the usual, the hazard rate, the mean residual life, and the likelihood ratio order. Recall that a family has the monotone likelihood ratio property if it has a likelihood ratio ordering. This suggests that, when the other parameters are known, there exists a test that is consistently the strongest for any one-sided hypothesis. According to Shaked and Shanthikumar [28], when two independent random variables, $W_1$ and $W_2$, have CDFs that are $F_{W_1}(w)$ and $F_{W_2}(w)$, respectively, $W_1$ is said to be smaller than $W_2$ in the

- Stochastic order ($W_1 \leq_{st} (W_2)$) if $F_{W_1}(w) \geq F_{W_2}(w) \ \forall w$
- Hazard rate order ($W_1 \leq_{hr} (W_2)$) if $h_{W_1}(w) \geq h_{W_2}(w) \forall w$
- Mean residual life order ($W_1 \leq_{mrl} (W_2)$) if $m_{W_1}(w) \geq m_{W_2}(w) \forall w$
- Likelihood ratio order ($W_1 \leq_{lr} (W_2)$) if $f_{W_1}(w)/f_{W_2}(w)$ decreases in w.

Assume that $W_i$, i = 1, 2 have the UPBXD with parameters $(a_i, b_i, \delta_i)$. Further, assume that $F_i(w)$ and $f_i(w)$ indicate, respectively, $W_i$'s CDF and PDF.

If $f_{W_1}(w)/f_{W_2}(w)$ is a decreasing function $\forall$ w, then, in terms of likelihood ratio order; $W_1$ is said to be stochastically less than $W_2$ ($W_1 \leq_{lr} W_2$)

Let $W_1 \sim$ UPBXD $(a_1, b_1, \delta_1)$ and $W_2 \sim$ UPBXD $(a_2, b_2, \delta_2)$, then the likelihood ratio ordering is as follows:

$$\frac{f_{W_1}(w)}{f_{W_2}(w)} = \frac{2a_1^2 b_1 \delta_1 (-\ln w)^{2b_1 - 1} e^{-(a_1(-\ln w)^{b_1})^2}\left[1 - \exp\left[-\left(a_1(-\ln w)^{b_1}\right)^2\right]\right]^{\delta_1}}{2a_2^2 b_2 \delta_2 (-\ln w)^{2b_2 - 1} e^{-(a_2(-\ln w)^{b_2})^2}\left[1 - \exp\left[-\left(a_2(-\ln w)^{b_2}\right)^2\right]\right]^{\delta_2}},$$

$$\frac{d}{dw} \log\left\{\frac{f_{W_1}(w)}{f_{W_2}(w)}\right\} = \frac{2(b_1 - b_2)}{w \ln w} + \frac{2\left[a_1^2 b_1 (-\ln w)^{2b_1 - 1} - a_2^2 b_2 (-\ln w)^{2b_2 - 1}\right]}{w} - \frac{2a_1^2 b_1 \delta_1 \exp - \left(a_1(-\ln w)^{b_1}\right)^2 (-\ln w)^{2b_1 - 1}}{w\left[1 - \exp\left[-\left(a_1(-\ln w)^{b_1}\right)^2\right]\right]^{\delta_1}}$$

$$+ \frac{2a_2^2 b_2 \delta_2 (-\ln w)^{2b_2 - 1} \exp - \left(a_2(-\ln w)^{b_2}\right)^2}{w\left[1 - \exp\left[\left(a_2(-\ln w)^{b_2}\right)^2\right]\right]^{\delta_2}}.$$

For $a_1 > a_2$, $b_1 > b_2$, $\delta_1 > \delta_2$, we get $\frac{d}{dw} \log\left[\frac{f_{W_1}(w)}{f_{W_2}(w)}\right] < 0$, for all $0 \leq w < 1$, hence $\frac{f_1(w)}{f_2(w)}$ is decreasing in w and hence $W_1 \leq_{lr} W_2$. Moreover, $W_1$ is said to be smaller than $W_2$ in other orderings such as SO ($W_1 \leq_{st} W_2$), HF($W_1 \leq_{hr} W_2$), and mean residual order ($W_1 \leq_{mrl} W_2$).

### 3.4. Stress–Stress Reliability

In statistical literature, the term "SS reliability" is used to characterize the reliability of a system subjected to random stress $W_2$ and having random strength $W_1$, with the system failing if $W_2$ is greater than $W_1$, that is; $R = P(W_2 < W_1)$. Let us assume that $W_1 \sim \text{UPBXD}\,(a_1, b, \delta_1)$ and $W_2 \sim \text{UPBXD}\,(a_2, b, \delta_2)$ are two independent random variables. The SS reliability of the UPBXD is then calculated as follows:

$$R = 2a_1 b \delta_1 \int\limits_0^1 w^{-1}(-\ln w)^{-2b-1} e^{-\left(a_1(-\ln w)^{-b}\right)^2}\left[1 - e^{-\left(a_1(-\ln w)^{-b}\right)^2}\right]^{\delta_1 - 1}\left[1 - \left[1 - e^{-\left(a_2(-\ln w)^{-b}\right)^2}\right]^{\delta_2}\right]dw. \quad (13)$$

Using the binomial expansions in (13), we get

$$
\begin{aligned}
R &= 1 - 2a_1 b \delta_1 \sum_{i_1,i_2=0}^{\infty} (-1)^{i_1+i_2} \binom{\delta_1 - 1}{i_1}\binom{\delta_2}{i_2} \int\limits_0^1 w^{-1}(-\ln w)^{-2b-1} e^{-\left(a_1^2 i_1 + i_2 a_2^2\right)(-\ln w)^{-2b}} dw \\
&= 1 - \sum_{k=0}^{\infty} \frac{(-1)^{i_1+i_2} a_1 \delta_1}{\left(a_1^2 i_1 + i_2 a_2^2\right)}\binom{\delta_1 - 1}{i_1}\binom{\delta_2}{i_2}.
\end{aligned}
\quad (14)
$$

As seen in (14) the SS reliability dependent on the parameters $a_1$, $a_2$, $\delta_1$, and $\delta_2$.

## 4. Parameter Estimation

The estimation methodologies for the parameters $(a, b, \delta)^T$ of the UPBXD are obtained in this part using Bayesian and non-Bayesian estimation approaches. We provide classical method for the UPBXD as ML and Bayesian estimation utilizing various loss functions, including the squared error loss function (SELF), the linear exponential (LINEX) loss function and entropy loss function (ELF).

### 4.1. Maximum Likelihood Method

Consider a population that has a UPBXD described by PDF (6) with an unknown parameter vector $\Im \equiv (a, b, \delta)$, and that a random sample of size $n$ is taken from that population. Following that, the likelihood of UPBXD for $\Im \equiv (a, b, \delta)$, say $L\left(w \middle| \Im\right)$, will be

$$L\left(w \middle| \Im\right) = 2^n a^{2n} b^n \delta^n e^{-\sum\limits_{i=1}^n (a(-\ln w_i)^b)^2} \prod_{i=1}^n \frac{(-\ln w_i)^{2b-1}}{w_i}\left[1 - \exp\left[-\left(a(-\ln w_i)^b\right)^2\right]\right]^{\delta - 1}. \quad (15)$$

The log likelihood function for $\Im \equiv (a, b, \delta)$, say $\updownarrow\left(w \middle| \Im\right)$, will be

$$
\begin{aligned}
\updownarrow\left(w \middle| \Im\right) &= n \ln(2\delta a^2 b) - \sum_{i=1}^n \ln w_i + (2b-1)\sum_{i=1}^n \ln(-\ln w_i) - a^2 \sum_{i=1}^n (-\ln w_i)^{2b} \\
&\quad + (\delta - 1)\sum_{i=1}^n \ln\left[1 - \exp\left[-\left(a(-\ln w_i)^b\right)^2\right]\right].
\end{aligned}
\quad (16)
$$

The nonlinear equations created by differentiating (16) with respect to $a, b,$ and $\delta$, are solved to obtain the ML estimator for the unknown parameters. The score vector components, say $U(\Im) = \frac{\partial \updownarrow(w|\Im)}{\partial \Im} = \left[\frac{\partial \updownarrow(w|\Im)}{\partial a}, \frac{\partial \updownarrow(w|\Im)}{\partial b}, \frac{\partial \updownarrow(w|\Im)}{\partial \delta}\right]^T$, are given by

$$U(a) = \frac{2n}{a} - 2a\sum_{i=1}^n (-\ln w_i)^{2b} + (\delta - 1)\sum_{i=1}^n \frac{2a(-\ln w_i)^{2b}}{\exp\left[\left(a(-\ln w_i)^b\right)^2\right] - 1}, \quad (17)$$

$$U(b) = \frac{n}{b} + 2\sum_{i=1}^n \ln(-\ln w_i) - 2a^2 \sum_{i=1}^n (-\ln w_i)^{2b}\ln(-\ln w_i) + \sum_{i=1}^n \frac{2a^2(\delta-1)(-\ln w_i)^{2b}\ln(-\ln w_i)}{\left[\exp\left(a^2(-\ln w_i)^{2b}\right) - 1\right]}, \quad (18)$$

$$U(\delta) = \frac{n}{\delta} + \sum_{i=1}^{n} \ln\left[ 1 - \exp\left[ -\left( a\left( -\ln w_i \right)^b \right)^2 \right] \right]. \tag{19}$$

The ML estimator of $\Im$, say $\hat{\Im}$, is achieved by solving the nonlinear system (17)–(19). These equations cannot be resolved analytically, but they can be resolved numerically by iterative statistical software techniques. We can use iterative methods, such as a Newton–Raphson algorithm, to obtain these estimates.

*4.2. Bayesian Estimation*

In this section, the Bayesian estimators based on different loss functions and associated highest posterior density (HPD) intervals of the UPBXD parameters are developed. The posterior distribution of $\Im$ is described in the following if we assume that the prior PDF of $\Im$ is unknown.

$$\pi(\Im | w) \propto L\left( w \big| \Im \right) g(\Im). \tag{20}$$

The posterior density of $\Im$ is defined in Equation (20) as $\pi(\Im | w)$, where on the right hand side $L\left( w \big| \Im \right)$ is the likelihood function of UPBXD $(\Im)$ and $g(\Im)$ is the prior density of $\Im$.

4.2.1. Prior Information

For the purpose of discussing Bayesian estimate, we assume that the parameters $a, b$, and $\delta$ are independently distributed using the gamma distribution. Let $q_j$ and $h_j$ where $j$ =1, 2, 3, be the scale and shape parameters for the gamma priors of $a, b$, and $\delta$. The following is a proportionate representation of the joint density of $a, b$, and $\delta$.

$$\pi(\Im) \propto a^{q_1-1} e^{-h_1 a} b^{q_2-1} e^{-h_2 b} \delta^{q_2-1} e^{-h_3 \delta}; q_j, h_j > 0, j = 1, 2, 3. \tag{21}$$

The hyper-parameters will be elicited using the informative priors. When $j = 1, \ldots, L$ and $k$ are the number of samples available from the UPBXD simulation, the mean and variance obtained using the ML estimates of the UPBXD $a, b$, and $\delta$ will be equal to the mean and variance of the considered priors (Gamma priors)$q_j$ and $h_j$. By equating $a, b$, and $\delta$ with the mean and variance of gamma priors, we may determine their respective means and variances. Thus, we obtain

$$q_1 = \frac{\left( \frac{1}{L} \sum_{j=1}^{L} \hat{a}^{\ j} \right)^2}{\frac{1}{L-1} \sum_{j=1}^{L} \left( \hat{a}^{\ j} - \frac{1}{L} \sum_{j=1}^{L} \hat{a}^{\ j} \right)^2}, \ \& \ h_1 = \frac{\frac{1}{L} \sum_{j=1}^{L} \hat{a}^{\ j}}{\frac{1}{L-1} \sum_{j=1}^{L} \left( \hat{a}^{\ j} - \frac{1}{L} \sum_{j=1}^{L} \hat{a}^{\ j} \right)^2},$$

$$q_2 = \frac{\left( \frac{1}{L} \sum_{j=1}^{L} \hat{b}^{\ j} \right)^2}{\frac{1}{L-1} \sum_{j=1}^{L} \left( \hat{b}^{\ j} - \frac{1}{L} \sum_{j=1}^{L} \hat{b}^{\ j} \right)^2}, \ \& \ h_2 = \frac{\frac{1}{L} \sum_{j=1}^{L} \hat{b}^{\ j}}{\frac{1}{L-1} \sum_{j=1}^{L} \left( \hat{b}^{\ j} - \frac{1}{L} \sum_{j=1}^{L} \hat{b}^{\ j} \right)^2},$$

$$q_3 = \frac{\left( \frac{1}{L} \sum_{j=1}^{L} \hat{\delta}^{\ j} \right)^2}{\frac{1}{L-1} \sum_{j=1}^{L} \left( \hat{\delta}^{\ j} - \frac{1}{L} \sum_{j=1}^{L} \hat{\delta}^{\ j} \right)^2}, \ \& \ h_3 = \frac{\frac{1}{L} \sum_{j=1}^{L} \hat{\delta}^{\ j}}{\frac{1}{L-1} \sum_{j=1}^{L} \left( \hat{\delta}^{\ j} - \frac{1}{L} \sum_{j=1}^{L} \hat{\delta}^{\ j} \right)^2}.$$

In regard to be solving the above two equations, the estimated hyper-parameters can be written as described in the following subsections.

4.2.2. Posterior Distribution

Here, the symmetric loss function (SELF), and asymmetric loss function (LINEX and ELF) are used to develop the Bayesian estimators for the same unknown parameters by utilizing independent gamma priors.

The likelihood function (15) and the joint prior function (21) are combined to form the joint posterior distribution. Hence, the joint posterior density function is

$$\pi(\Im|w) \propto a^{2n+q_1-1} e^{-h_1 a} b^{n+q_2-1} e^{-h_2 b} \delta^{n+q_2-1} e^{-h_3\delta} e^{-\sum\limits_{i=1}^{n} (a(-\ln w_i)^b)^2} \prod_{i=1}^{n} \frac{(-\ln w_i)^{2b-1}}{w_i} \left[1 - \exp\left[-\left(a(-\ln w_i)^b\right)^2\right]\right]^{\delta-1}. \tag{22}$$

The SELF, is defined as follows:

$$\text{SELF}\left(\widetilde{\Im}, \Im\right) \propto \left(\widetilde{\Im} - \Im\right)^2.$$

The Bayesian estimator of $\Im$ under SELF is as follows:

$$\widetilde{\Im} = \int\limits_0^\infty \int\limits_0^\infty \int\limits_0^\infty \Im \pi(\Im|\underline{w}) \, da \, db \, d\delta. \tag{23}$$

The LINEX, as asymmetric loss function, which is denoted by $\widetilde{\widetilde{\Im}}$, is the derived as follows:

$$\text{Linex}\left(\widetilde{\widetilde{\Im}}, \Im\right) \propto e^{c(\Im-\Im)} - c(\Im - \Im) - 1; \quad c \neq 0.$$

The Bayesian estimator of $\Im$ under LINEX loss function is as follows:

$$\widetilde{\widetilde{\Im}} = \int\limits_0^\infty \int\limits_0^\infty \int\limits_0^\infty e^{-c\Im} \pi(\Im|\underline{w}) \, da \, db \, d\delta. \tag{24}$$

The ELF was first suggested by James and Stein [29] to estimate the Variance–Covariance (i.e., dispersion) matrix of the multivariate normal distribution. According to Calabria and Pulcini [30], the ELF is an excellent asymmetric loss function. The form's ELF is thought of as

$$\text{ELF}\left(\breve{\Im}, \Im\right) \propto \frac{c^2}{2}\left[\ln\left(\breve{\Im}\right) - \ln(\Im)\right]; \quad c \neq 0.$$

The Bayesian estimator of $\Im$ under ELF is as follows:

$$\breve{\Im} = \int\limits_0^\infty \int\limits_0^\infty \int\limits_0^\infty \Im^{-c} \pi(\Im|\underline{w}) \, da \, db \, d\delta. \tag{25}$$

The Bayes estimator of $a, b$, and $\delta$ via different loss functions cannot be expressed in an explicit statement, as is evident from Equations (23)–(25). To do this, we suggest generating samples from conditional posterior distribution using Bayes Monte Carlo Markov chain (MCMC) techniques in order to compute the acquired Bayes estimates and create associated HPD intervals.

### 4.2.3. Markov Chain Monte Carlo

Since it is challenging to solve these integrals analytically, the MCMC method will be used. The most important sub-classes of MCMC algorithms are Gibbs sampling and the Metropolis–Hastings (MH) samplers. To do this, it regards a candidate value produced from a proposal distribution as normal for each iteration of the process, the MH method is comparable to acceptance–rejection sampling. From Equation (22), the full conditional density of $a, b$, and $\delta$ are provided, respectively, to execute the MCMC sampler as follows:

$$\pi(a|b,\delta,w) \propto a^{2n+q_1-1} e^{-w_1 a} e^{-\sum\limits_{i=1}^{n} (a(-\ln w_i)^b)^2} \prod_{i=1}^{n} \left[1 - \exp\left[-\left(a(-\ln w_i)^b\right)^2\right]\right]^{\delta-1},$$

$$\pi(b|a,\delta,w) \propto b^{n+q_2-1} e^{-w_2 b} e^{-\sum\limits_{i=1}^{n} (a(-\ln w_i)^b)^2} \prod_{i=1}^{n} \frac{(-\ln w_i)^{2b-1}}{w_i} \left[1 - \exp\left[-\left(a(-\ln w_i)^b\right)^2\right]\right]^{\delta-1},$$

and

$$\pi(\delta|a,b,w) \propto \delta^{n+q_2-1} e^{-w_2\delta} \prod_{i=1}^{n} \left[ 1 - \exp\left[ -\left( a\left( -\ln w_i\right)^b\right)^2\right]\right]^{\delta-1}.$$

It is thought that the MH algorithm can resolve this issue (for detail, see Alrumayh et al. [31] and Almetwally et al. [32]). The MH algorithm's sampling procedure is carried out as follows:

**Step 1:** Set the initial values $a^{(0)} = \hat{a}$, $b^{(0)} = \hat{b}$, and $\delta^{(0)} = \hat{\delta}$.

**Step 2:** $Set I = 1$.

**Step 3:** Generate $a^*$, $b^*$ and $\delta^*$ from $N(\hat{a}, V_{\hat{a}})$, $N(\hat{b}, V_{\hat{b}})$ and $N(\hat{\delta}, V_{\hat{\delta}})$, respectively.

**Step 4:** Obtain $\hbar_a = \min\left[1, \frac{\pi\left(a^*|b^{(I-1)},\delta^{(I-1)},\underline{w}\right)}{\pi\left(a^{(I-1)}|b^{(I-1)},\delta^{(I-1)},\underline{w}\right)}\right]$,

$\hbar_b = \min\left[1, \frac{\pi\left(b^*|a^{(I-1)},\delta^{(I-1)},\underline{w}\right)}{\pi\left(b^{(I-1)}|a^{(I-1)},\delta^{(I-1)},\underline{w}\right)}\right]$, $\hbar_\delta = \min\left[1, \frac{\pi\left(\delta^*|a^{(I-1)},b^{(I-1)},\underline{w}\right)}{\pi\left(\delta^{(I-1)}|a^{(I-1)},b^{(I-1)},\underline{w}\right)}\right]$.

**Step 5:** Generate samples $U_j$ $j$ =1,2,3 from the uniform $U(0, 1)$ distribution.

**Step 6:** If $U_1 \leq \hbar_a, U_2 \leq \hbar_b$, and $U_3 \leq \hbar_\delta$, then set $a^{(I)} = a^*, b^{(I)} = b^*$, $\delta^{(I)} = \delta^*$; otherwise $a^{(I)} = a^{(I-1)}, b^{(I)} = b^{(I-1)}$, and $\delta^{(I)} = \delta^{(I-1)}$.

**Step 7:** Set $I = I+1$.

**Step 8:** Repeat steps 3–7 $B$ times and obtain $a^{(I)}, b^{(I)}$, and $\delta^{(I)}$, for $I$ = 1, 2,..., B.

### 4.2.4. Highest Posterior Density Interval

Using the technique suggested by Chen and Shao [33], $100(1-\alpha)\%$ HPD interval estimates of $a, b$, and $\delta$ are created. The MCMC samples of $\Im^{(j)}$ for $j$ = 1, ... , B are first ordered. Therefore, the two-sided $100(1-\alpha)\%$ HPD interval of $\Im$ is given by

$$\left(a^{\left[\frac{\alpha}{2}B\right]}, a^{\left[(1-\frac{\alpha}{2})B\right]}\right), \left(b^{\left[\frac{\alpha}{2}B\right]}, b^{\left[(1-\frac{\alpha}{2})B\right]}\right) \text{ and } \left(\delta^{\left[\frac{\alpha}{2}B\right]}, \delta^{\left[(1-\frac{\alpha}{2})B\right]}\right),$$

where $a^{[1]} < a^{[2]} < \ldots < a^{[B]}$, $b^{[1]} < b^{[2]} < \ldots < b^{[B]}$, and $\delta^{[1]} < \delta^{[2]} < \ldots < \delta^{[B]}$.

## 5. Simulation

A Monte Carlo simulation was run to evaluate the performance of the proposed point and interval estimators that were introduced in the previous sections. Based on various selections for sample size $n$ as 40, 80, and 160, UPBXD was used to create a total of 5000 samples. To compare the results of Bayesian estimate based on various loss functions, the bias and mean squared errors (MSE) were calculated. The UPBXD was used to generate the data for the lifetime of various parameters $a, b$, and $\delta$, as follows.

In Table 3: $a$ = 0.5, $b$ = 0.6 and $\delta$ = 0.5, 1.2 and 3. In Table 4: $a$ = 0.5, $b$ = 1.7 and $\delta$ = 0.5, 1.2 and 3. In Table 5: $a$ = 2, $b$ = 0.6 and $\delta$ = 0.5, 1.2 and 3. In Table 6: $a$ = 2, $b$ = 1.7 and $\delta$ = 0.5, 1.2 and 3.

The hybrid MCMC algorithm described in Section 4.2.3 was adopted to generate 12,000 MCMC samples, and we discarded the first 2000 values as 'burn-in'. Accordingly, the 10,000 MCMC samples were used to produce the average Bayes MCMC estimates and 95% two-sided Bayesian credible intervals.

1. Algorithm for simulation: By establishing all simulation controls, we can build our model. The following actions must be finished in this stage in the correct order:
2. Assume different values for the UPBXD parameter vector and sample size.
3. Make the sample random values for the UPBXD using uniform and the QF in Equation (7).
4. We calculated the accuracy measures for each Bayes estimates of the UPBXD parameters using MH algorithm.
5. This experiment should be run (L-1) times.

**Table 3.** Bayesian inference with different loss functions when $a = 0.5$, $b = 0.6$.

| $\delta$ | $n$ | $a=0.5,$ $b=0.6$ | SELF | | | LINEX ($c = -1.5$) | | | LINEX ($c = 1.5$) | | | ELF ($c = -1.5$) | | | ELF ($c = 1.5$) | | |
|---|---|---|---|---|---|---|---|---|---|---|---|---|---|---|---|---|---|
| | | | Bias | MSE | LCCI | Bias | MSE | LCCI | Bias | MSE | LCCI | Bias | MSE | LCCI | Bias | MSE | LCCI |
| | 40 | $a$ | 0.1392 | 0.0224 | 0.2075 | 0.1448 | 0.0242 | 0.2140 | 0.1335 | 0.0207 | 0.2021 | 0.1422 | 0.0233 | 0.2098 | 0.1237 | 0.0181 | 0.1985 |
| | | $b$ | −0.0261 | 0.0008 | 0.0423 | −0.0253 | 0.0008 | 0.0420 | −0.0269 | 0.0007 | 0.0427 | −0.0256 | 0.0008 | 0.0421 | −0.0284 | 0.0007 | 0.0426 |
| | | $\delta$ | 0.6123 | 0.4095 | 0.7079 | 0.6587 | 0.4793 | 0.7990 | 0.5592 | 0.3364 | 0.5954 | 0.6267 | 0.4298 | 0.7275 | 0.5222 | 0.2932 | 0.5536 |
| 0.5 | 80 | $a$ | 0.0900 | 0.0094 | 0.1390 | 0.0927 | 0.0099 | 0.1413 | 0.0872 | 0.0088 | 0.1365 | 0.0915 | 0.0097 | 0.1400 | 0.0818 | 0.0079 | 0.1331 |
| | | $b$ | −0.0136 | 0.0002 | 0.0262 | −0.0133 | 0.0002 | 0.0262 | −0.0139 | 0.0002 | 0.0265 | −0.0134 | 0.0002 | 0.0263 | −0.0144 | 0.0003 | 0.0267 |
| | | $\delta$ | 0.4229 | 0.1878 | 0.3562 | 0.4479 | 0.2117 | 0.3996 | 0.3954 | 0.1632 | 0.3134 | 0.4323 | 0.1965 | 0.3721 | 0.3672 | 0.1405 | 0.2908 |
| | 160 | $a$ | 0.0843 | 0.0078 | 0.1017 | 0.0865 | 0.0082 | 0.1032 | 0.0821 | 0.0074 | 0.0994 | 0.0856 | 0.0080 | 0.1026 | 0.0779 | 0.0067 | 0.0967 |
| | | $b$ | −0.0135 | 0.0002 | 0.0237 | −0.0126 | 0.0002 | 0.0237 | −0.0127 | 0.0002 | 0.0238 | −0.0136 | 0.0002 | 0.0237 | −0.0137 | 0.0002 | 0.0241 |
| | | $\delta$ | 0.3906 | 0.1567 | 0.2496 | 0.4121 | 0.1751 | 0.2732 | 0.3670 | 0.1380 | 0.2186 | 0.3991 | 0.1638 | 0.2564 | 0.3411 | 0.1190 | 0.2017 |
| | 40 | $a$ | 0.0804 | 0.0082 | 0.1604 | 0.0841 | 0.0088 | 0.1614 | 0.0767 | 0.0075 | 0.1585 | 0.0825 | 0.0085 | 0.1609 | 0.0695 | 0.0065 | 0.1600 |
| | | $b$ | −0.0204 | 0.0008 | 0.0769 | −0.0187 | 0.0008 | 0.0775 | −0.0221 | 0.0009 | 0.0757 | −0.0195 | 0.0008 | 0.0772 | −0.0252 | 0.0010 | 0.0756 |
| | | $\delta$ | 0.6959 | 0.5321 | 0.8034 | 0.7936 | 0.6999 | 0.9837 | 0.5905 | 0.3779 | 0.6559 | 0.7138 | 0.5604 | 0.8249 | 0.5986 | 0.3913 | 0.7033 |
| 1.2 | 80 | $a$ | 0.0417 | 0.0025 | 0.1130 | 0.0431 | 0.0027 | 0.1135 | 0.0403 | 0.0024 | 0.1112 | 0.0426 | 0.0026 | 0.1131 | 0.0372 | 0.0022 | 0.1105 |
| | | $b$ | −0.0076 | 0.0002 | 0.0494 | −0.0069 | 0.0002 | 0.0498 | −0.0082 | 0.0002 | 0.0491 | −0.0072 | 0.0002 | 0.0496 | −0.0094 | 0.0003 | 0.0486 |
| | | $\delta$ | 0.3724 | 0.1515 | 0.4317 | 0.4044 | 0.1794 | 0.4884 | 0.3394 | 0.1252 | 0.3813 | 0.3792 | 0.1571 | 0.4387 | 0.3370 | 0.1238 | 0.3846 |
| | 160 | $a$ | 0.0371 | 0.0018 | 0.0780 | 0.0380 | 0.0018 | 0.0789 | 0.0363 | 0.0017 | 0.0774 | 0.0377 | 0.0018 | 0.0785 | 0.0344 | 0.0016 | 0.0769 |
| | | $b$ | −0.0082 | 0.0002 | 0.0370 | −0.0079 | 0.0002 | 0.0371 | −0.0086 | 0.0002 | 0.0370 | −0.0080 | 0.0002 | 0.0371 | −0.0092 | 0.0002 | 0.0371 |
| | | $\delta$ | 0.3429 | 0.1242 | 0.3101 | 0.3689 | 0.1443 | 0.3453 | 0.3161 | 0.1050 | 0.2732 | 0.3486 | 0.1284 | 0.3168 | 0.3135 | 0.1034 | 0.2735 |
| | 40 | $a$ | 0.0224 | 0.0019 | 0.1450 | 0.0249 | 0.0020 | 0.1457 | 0.0198 | 0.0018 | 0.1438 | 0.0240 | 0.0020 | 0.1448 | 0.0143 | 0.0016 | 0.1431 |
| | | $b$ | −0.0036 | 0.0011 | 0.1197 | −0.0013 | 0.0011 | 0.1215 | −0.0059 | 0.0011 | 0.1188 | −0.0023 | 0.0011 | 0.1199 | −0.0101 | 0.0011 | 0.1184 |
| | | $\delta$ | 0.4244 | 0.2485 | 1.0159 | 0.4897 | 0.3280 | 1.1489 | 0.3576 | 0.1793 | 0.8598 | 0.4307 | 0.2556 | 1.0306 | 0.3919 | 0.2142 | 0.9512 |
| 3 | 80 | $a$ | 0.0091 | 0.0008 | 0.1013 | 0.0101 | 0.0008 | 0.1021 | 0.0080 | 0.0007 | 0.1011 | 0.0098 | 0.0008 | 0.1019 | 0.0057 | 0.0007 | 0.1005 |
| | | $b$ | −0.0010 | 0.0005 | 0.0904 | 0.0000 | 0.0005 | 0.0907 | −0.0019 | 0.0005 | 0.0898 | −0.0004 | 0.0005 | 0.0906 | −0.0037 | 0.0005 | 0.0899 |
| | | $\delta$ | 0.1950 | 0.0567 | 0.5279 | 0.2113 | 0.0661 | 0.5690 | 0.1786 | 0.0481 | 0.4905 | 0.1967 | 0.0577 | 0.5322 | 0.1866 | 0.0523 | 0.5095 |
| | 160 | $a$ | 0.0086 | 0.0005 | 0.0767 | 0.0091 | 0.0005 | 0.0766 | 0.0080 | 0.0004 | 0.0757 | 0.0089 | 0.0005 | 0.0766 | 0.0067 | 0.0004 | 0.0753 |
| | | $b$ | −0.0016 | 0.0004 | 0.0789 | −0.0010 | 0.0004 | 0.0791 | −0.0022 | 0.0004 | 0.0783 | −0.0013 | 0.0004 | 0.0791 | −0.0032 | 0.0004 | 0.0791 |
| | | $\delta$ | 0.1804 | 0.0409 | 0.3497 | 0.1911 | 0.0459 | 0.3734 | 0.1696 | 0.0361 | 0.3294 | 0.1815 | 0.0414 | 0.3519 | 0.1748 | 0.0384 | 0.3404 |

**Table 4.** Bayesian inference with different loss functions when $a = 0.5$, $b = 1.7$.

| $\delta$ | $a = 0.5$, $b = 1.7$ n | | SELF Bias | MSE | LCCI | LINEX ($c = -1.5$) Bias | MSE | LCCI | LINEX ($c = 1.5$) Bias | MSE | LCCI | ELF ($c = -1.5$) Bias | MSE | LCCI | ELF ($c = 1.5$) Bias | MSE | LCCI |
|---|---|---|---|---|---|---|---|---|---|---|---|---|---|---|---|---|---|
| | | $a$ | 0.1212 | 0.0179 | 0.2155 | 0.1257 | 0.0191 | 0.2199 | 0.1167 | 0.0167 | 0.2110 | 0.1236 | 0.0185 | 0.2172 | 0.1086 | 0.0147 | 0.2073 |
| | 40 | $b$ | −0.0308 | 0.0013 | 0.0722 | −0.0281 | 0.0012 | 0.0716 | −0.0336 | 0.0015 | 0.0742 | −0.0303 | 0.0013 | 0.0722 | −0.0336 | 0.0015 | 0.0744 |
| | | $\delta$ | 0.5944 | 0.3901 | 0.7286 | 0.6374 | 0.4534 | 0.8182 | 0.5457 | 0.3238 | 0.6240 | 0.6078 | 0.4086 | 0.7470 | 0.5113 | 0.2844 | 0.5813 |
| | | $a$ | 0.0845 | 0.0084 | 0.1382 | 0.0870 | 0.0089 | 0.1402 | 0.0820 | 0.0079 | 0.1365 | 0.0859 | 0.0087 | 0.1386 | 0.0771 | 0.0071 | 0.1335 |
| 0.5 | 80 | $b$ | −0.0229 | 0.0009 | 0.0662 | −0.0215 | 0.0008 | 0.0663 | −0.0242 | 0.0010 | 0.0661 | −0.0226 | 0.0009 | 0.0662 | −0.0242 | 0.0010 | 0.0661 |
| | | $\delta$ | 0.4267 | 0.1911 | 0.3714 | 0.4511 | 0.2147 | 0.4076 | 0.3995 | 0.1666 | 0.3243 | 0.4359 | 0.1996 | 0.3810 | 0.3717 | 0.1439 | 0.2986 |
| | | $a$ | 0.0754 | 0.0062 | 0.0897 | 0.0771 | 0.0065 | 0.0910 | 0.0736 | 0.0059 | 0.0889 | 0.0764 | 0.0064 | 0.0904 | 0.0701 | 0.0054 | 0.0877 |
| | 160 | $b$ | −0.0196 | 0.0005 | 0.0476 | −0.0189 | 0.0005 | 0.0477 | −0.0203 | 0.0006 | 0.0476 | −0.0195 | 0.0005 | 0.0477 | −0.0203 | 0.0006 | 0.0476 |
| | | $\delta$ | 0.3890 | 0.1550 | 0.2339 | 0.4099 | 0.1725 | 0.2573 | 0.3660 | 0.1368 | 0.2117 | 0.3973 | 0.1618 | 0.2414 | 0.3404 | 0.1182 | 0.1901 |
| | | $a$ | 0.0722 | 0.0069 | 0.1623 | 0.0751 | 0.0074 | 0.1645 | 0.0693 | 0.0065 | 0.1600 | 0.0739 | 0.0072 | 0.1631 | 0.0635 | 0.0057 | 0.1576 |
| | 40 | $b$ | −0.0311 | 0.0033 | 0.1871 | −0.0241 | 0.0030 | 0.1885 | −0.0380 | 0.0037 | 0.1841 | −0.0297 | 0.0032 | 0.1865 | −0.0380 | 0.0037 | 0.1853 |
| | | $\delta$ | 0.7333 | 0.5923 | 0.8827 | 0.8377 | 0.7792 | 1.0665 | 0.6187 | 0.4166 | 0.7014 | 0.7523 | 0.6236 | 0.9119 | 0.6288 | 0.4342 | 0.7495 |
| | | $a$ | 0.0423 | 0.0026 | 0.1102 | 0.0436 | 0.0027 | 0.1111 | 0.0410 | 0.0025 | 0.1094 | 0.0431 | 0.0026 | 0.1108 | 0.0383 | 0.0022 | 0.1089 |
| 1.2 | 80 | $b$ | −0.0179 | 0.0020 | 0.1693 | −0.0152 | 0.0020 | 0.1689 | −0.0207 | 0.0021 | 0.1678 | −0.0174 | 0.0020 | 0.1694 | −0.0207 | 0.0022 | 0.1681 |
| | | $\delta$ | 0.3962 | 0.1709 | 0.4626 | 0.4310 | 0.2032 | 0.5201 | 0.3599 | 0.1403 | 0.4013 | 0.4036 | 0.1774 | 0.4722 | 0.3577 | 0.1390 | 0.4080 |
| | | $a$ | 0.0365 | 0.0018 | 0.0839 | 0.0373 | 0.0019 | 0.0847 | 0.0357 | 0.0017 | 0.0831 | 0.0370 | 0.0018 | 0.0845 | 0.0340 | 0.0016 | 0.0822 |
| | 160 | $b$ | −0.0163 | 0.0017 | 0.1472 | −0.0146 | 0.0016 | 0.1465 | −0.0180 | 0.0018 | 0.1486 | −0.0160 | 0.0017 | 0.1468 | −0.0180 | 0.0018 | 0.1488 |
| | | $\delta$ | 0.3476 | 0.1271 | 0.3037 | 0.3738 | 0.1475 | 0.3367 | 0.3205 | 0.1076 | 0.2707 | 0.3533 | 0.1314 | 0.3102 | 0.3179 | 0.1060 | 0.2717 |
| | | $a$ | 0.0202 | 0.0018 | 0.1484 | 0.0222 | 0.0019 | 0.1503 | 0.0182 | 0.0017 | 0.1483 | 0.0215 | 0.0019 | 0.1493 | 0.0138 | 0.0016 | 0.1477 |
| | 40 | $b$ | −0.0037 | 0.0100 | 0.3923 | 0.0087 | 0.0104 | 0.3968 | −0.0161 | 0.0100 | 0.3886 | −0.0013 | 0.0100 | 0.3898 | −0.0160 | 0.0102 | 0.3926 |
| | | $\delta$ | 0.4503 | 0.2804 | 1.0727 | 0.5244 | 0.3753 | 1.2260 | 0.3748 | 0.1981 | 0.9209 | 0.4575 | 0.2888 | 1.0900 | 0.4138 | 0.2398 | 1.0071 |
| | | $a$ | 0.0106 | 0.0008 | 0.1023 | 0.0113 | 0.0008 | 0.1019 | 0.0098 | 0.0008 | 0.1018 | 0.0110 | 0.0008 | 0.1019 | 0.0081 | 0.0008 | 0.1018 |
| 3 | 80 | $b$ | −0.0056 | 0.0052 | 0.2814 | −0.0015 | 0.0052 | 0.2815 | −0.0096 | 0.0053 | 0.2807 | −0.0048 | 0.0052 | 0.2808 | −0.0096 | 0.0053 | 0.2821 |
| | | $\delta$ | 0.2074 | 0.0618 | 0.5202 | 0.2251 | 0.0722 | 0.5520 | 0.1897 | 0.0522 | 0.4859 | 0.2092 | 0.0628 | 0.5225 | 0.1983 | 0.0568 | 0.5058 |
| | | $a$ | 0.0082 | 0.0004 | 0.0708 | 0.0085 | 0.0004 | 0.0709 | 0.0078 | 0.0004 | 0.0706 | 0.0084 | 0.0004 | 0.0709 | 0.0069 | 0.0004 | 0.0702 |
| | 160 | $b$ | −0.0037 | 0.0025 | 0.1941 | −0.0018 | 0.0025 | 0.1921 | −0.0056 | 0.0025 | 0.1949 | −0.0033 | 0.0025 | 0.1941 | −0.0056 | 0.0025 | 0.1953 |
| | | $\delta$ | 0.1895 | 0.0457 | 0.3920 | 0.2014 | 0.0517 | 0.4220 | 0.1775 | 0.0400 | 0.3662 | 0.1908 | 0.0463 | 0.3952 | 0.1833 | 0.0428 | 0.3771 |

**Table 5.** Bayesian inference with different loss functions when $a = 2$, $b = 0.6$.

| $\delta$ | n | $a = 2, b = 0.6$ | SELF | | | LINEX ($c = -1.5$) | | | LINEX ($c = 1.5$) | | | ELF ($c = -1.5$) | | | ELF ($c = 1.5$) | | |
|---|---|---|---|---|---|---|---|---|---|---|---|---|---|---|---|---|---|
| | | | Bias | MSE | LCCI | Bias | MSE | LCCI | Bias | MSE | LCCI | Bias | MSE | LCCI | Bias | MSE | LCCI |
| 0.5 | 40 | $a$ | 0.1442 | 0.0330 | 0.4135 | 0.1607 | 0.0393 | 0.4250 | 0.1275 | 0.0273 | 0.3941 | 0.1468 | 0.0339 | 0.4181 | 0.1313 | 0.0286 | 0.4004 |
| | | $b$ | −0.0388 | 0.0021 | 0.0635 | −0.0380 | 0.0020 | 0.0621 | −0.0396 | 0.0021 | 0.0652 | −0.0382 | 0.0020 | 0.0625 | −0.0411 | 0.0023 | 0.0681 |
| | | $\delta$ | 0.5679 | 0.3597 | 0.7317 | 0.6074 | 0.4169 | 0.8179 | 0.5232 | 0.2998 | 0.6075 | 0.5804 | 0.3765 | 0.7545 | 0.4907 | 0.2634 | 0.5613 |
| | 80 | $a$ | 0.0678 | 0.0096 | 0.2757 | 0.0735 | 0.0107 | 0.2852 | 0.0622 | 0.0087 | 0.2721 | 0.0687 | 0.0098 | 0.2764 | 0.0634 | 0.0089 | 0.2724 |
| | | $b$ | −0.0275 | 0.0009 | 0.0399 | −0.0271 | 0.0008 | 0.0395 | −0.0279 | 0.0009 | 0.0402 | −0.0272 | 0.0009 | 0.0397 | −0.0286 | 0.0009 | 0.0410 |
| | | $\delta$ | 0.3992 | 0.1676 | 0.3463 | 0.4199 | 0.1861 | 0.3752 | 0.3764 | 0.1482 | 0.3121 | 0.4073 | 0.1745 | 0.3559 | 0.3518 | 0.1293 | 0.2869 |
| | 160 | $a$ | 0.0582 | 0.0062 | 0.2080 | 0.0614 | 0.0067 | 0.2116 | 0.0550 | 0.0058 | 0.2065 | 0.0587 | 0.0063 | 0.2084 | 0.0556 | 0.0058 | 0.2069 |
| | | $b$ | −0.0282 | 0.0008 | 0.0342 | −0.0279 | 0.0009 | 0.0342 | −0.0285 | 0.0010 | 0.0341 | −0.0280 | 0.0009 | 0.0334 | −0.0291 | 0.0010 | 0.0332 |
| | | $\delta$ | 0.3702 | 0.1410 | 0.2369 | 0.3886 | 0.1557 | 0.2640 | 0.3501 | 0.1257 | 0.2096 | 0.3776 | 0.1468 | 0.2473 | 0.3270 | 0.1094 | 0.1937 |
| 1.2 | 40 | $a$ | 0.0995 | 0.0178 | 0.3423 | 0.1126 | 0.0210 | 0.3483 | 0.0866 | 0.0150 | 0.3373 | 0.1016 | 0.0182 | 0.3429 | 0.0892 | 0.0156 | 0.3408 |
| | | $b$ | −0.0373 | 0.0104 | 0.0748 | −0.0353 | 0.0132 | 0.0747 | −0.0400 | 0.0051 | 0.0744 | −0.0363 | 0.0109 | 0.0746 | −0.0429 | 0.0062 | 0.0747 |
| | | $\delta$ | 0.7256 | 0.5834 | 0.9188 | 0.8305 | 0.7724 | 1.1304 | 0.6121 | 0.4104 | 0.7213 | 0.7446 | 0.6146 | 0.9436 | 0.6218 | 0.4277 | 0.7719 |
| | 80 | $a$ | 0.0395 | 0.0057 | 0.2532 | 0.0438 | 0.0062 | 0.2577 | 0.0352 | 0.0053 | 0.2509 | 0.0402 | 0.0058 | 0.2542 | 0.0360 | 0.0054 | 0.2511 |
| | | $b$ | −0.0198 | 0.0006 | 0.0506 | −0.0191 | 0.0005 | 0.0511 | −0.0205 | 0.0006 | 0.0497 | −0.0194 | 0.0005 | 0.0509 | −0.0218 | 0.0006 | 0.0492 |
| | | $\delta$ | 0.3838 | 0.1612 | 0.4450 | 0.4172 | 0.1913 | 0.4972 | 0.3493 | 0.1327 | 0.3917 | 0.3909 | 0.1672 | 0.4565 | 0.3470 | 0.1313 | 0.3945 |
| | 160 | $a$ | 0.0352 | 0.0035 | 0.1802 | 0.0375 | 0.0037 | 0.1816 | 0.0329 | 0.0033 | 0.1780 | 0.0356 | 0.0036 | 0.1805 | 0.0334 | 0.0034 | 0.1788 |
| | | $b$ | −0.0194 | 0.0006 | 0.0366 | −0.0191 | 0.0007 | 0.0365 | −0.0198 | 0.0008 | 0.0366 | −0.0192 | 0.0007 | 0.0365 | −0.0206 | 0.0009 | 0.0368 |
| | | $\delta$ | 0.3437 | 0.1244 | 0.2940 | 0.3693 | 0.1441 | 0.3334 | 0.3172 | 0.1055 | 0.2635 | 0.3493 | 0.1285 | 0.3026 | 0.3146 | 0.1039 | 0.2647 |
| 3 | 40 | $a$ | 0.0382 | 0.0061 | 0.2646 | 0.0461 | 0.0069 | 0.2666 | 0.0304 | 0.0054 | 0.2582 | 0.0395 | 0.0062 | 0.2640 | 0.0318 | 0.0055 | 0.2594 |
| | | $b$ | −0.0038 | 0.0167 | 0.0996 | −0.0002 | 0.0318 | 0.1001 | −0.0081 | 0.0036 | 0.0979 | −0.0025 | 0.0187 | 0.0993 | −0.0113 | 0.0051 | 0.0978 |
| | | $\delta$ | 0.4644 | 0.2952 | 1.0815 | 0.5388 | 0.3941 | 1.2013 | 0.3883 | 0.2092 | 0.9174 | 0.4716 | 0.3039 | 1.0866 | 0.4279 | 0.2530 | 1.0157 |
| | 80 | $a$ | 0.0103 | 0.0031 | 0.2140 | 0.0134 | 0.0032 | 0.2145 | 0.0071 | 0.0030 | 0.2110 | 0.0108 | 0.0031 | 0.2143 | 0.0077 | 0.0030 | 0.2116 |
| | | $b$ | −0.0058 | 0.0005 | 0.0801 | −0.0047 | 0.0005 | 0.0800 | −0.0068 | 0.0005 | 0.0796 | −0.0052 | 0.0005 | 0.0802 | −0.0086 | 0.0005 | 0.0795 |
| | | $\delta$ | 0.2169 | 0.0675 | 0.5525 | 0.2362 | 0.0797 | 0.5974 | 0.1974 | 0.0564 | 0.5100 | 0.2188 | 0.0687 | 0.5577 | 0.2069 | 0.0618 | 0.5310 |
| | 160 | $a$ | 0.0115 | 0.0014 | 0.1356 | 0.0129 | 0.0014 | 0.1365 | 0.0100 | 0.0013 | 0.1352 | 0.0117 | 0.0014 | 0.1357 | 0.0102 | 0.0013 | 0.1353 |
| | | $b$ | −0.0051 | 0.0002 | 0.0534 | −0.0046 | 0.0002 | 0.0532 | −0.0055 | 0.0002 | 0.0531 | −0.0048 | 0.0002 | 0.0533 | −0.0063 | 0.0002 | 0.0529 |
| | | $\delta$ | 0.1894 | 0.0461 | 0.3862 | 0.2014 | 0.0521 | 0.4096 | 0.1773 | 0.0404 | 0.3600 | 0.1907 | 0.0467 | 0.3896 | 0.1832 | 0.0431 | 0.3755 |

**Table 6.** Bayesian inference with different loss functions when $a = 2$, $b = 1.7$.

| $\delta$ | $n$ | $a=2, b=1.7$ | SELF | | | LINEX ($c=-1.5$) | | | LINEX ($c=1.5$) | | | ELF ($c=-1.5$) | | | ELF ($c=1.5$) | | |
|---|---|---|---|---|---|---|---|---|---|---|---|---|---|---|---|---|---|
| | | | Bias | MSE | LCCI | Bias | MSE | LCCI | Bias | MSE | LCCI | Bias | MSE | LCCI | Bias | MSE | LCCI |
| | | $a$ | 0.1255 | 0.0232 | 0.3228 | 0.1388 | 0.0274 | 0.3337 | 0.1122 | 0.0194 | 0.3150 | 0.1276 | 0.0238 | 0.3237 | 0.1151 | 0.0203 | 0.3166 |
| | 40 | $b$ | −0.0538 | 0.0036 | 0.0995 | −0.0507 | 0.0032 | 0.0975 | −0.0568 | 0.0040 | 0.1018 | −0.0532 | 0.0035 | 0.0990 | −0.0569 | 0.0040 | 0.1021 |
| | | $\delta$ | 0.5381 | 0.3172 | 0.6043 | 0.5710 | 0.3603 | 0.6771 | 0.5012 | 0.2719 | 0.5360 | 0.5489 | 0.3305 | 0.6253 | 0.4723 | 0.2413 | 0.5051 |
| | | $a$ | 0.0715 | 0.0094 | 0.2549 | 0.0774 | 0.0105 | 0.2621 | 0.0657 | 0.0084 | 0.2483 | 0.0724 | 0.0096 | 0.2567 | 0.0668 | 0.0086 | 0.2506 |
| 0.5 | 80 | $b$ | −0.0505 | 0.0032 | 0.0934 | −0.0485 | 0.0030 | 0.0927 | −0.0525 | 0.0034 | 0.0944 | −0.0501 | 0.0032 | 0.0932 | −0.0526 | 0.0034 | 0.0946 |
| | | $\delta$ | 0.4003 | 0.1681 | 0.3332 | 0.4211 | 0.1868 | 0.3669 | 0.3775 | 0.1487 | 0.3008 | 0.4083 | 0.1751 | 0.3439 | 0.3531 | 0.1297 | 0.2722 |
| | | $a$ | 0.0580 | 0.0054 | 0.1784 | 0.0608 | 0.0058 | 0.1805 | 0.0551 | 0.0050 | 0.1763 | 0.0584 | 0.0055 | 0.1784 | 0.0557 | 0.0051 | 0.1770 |
| | 160 | $b$ | −0.0364 | 0.0016 | 0.0611 | −0.0355 | 0.0015 | 0.0604 | −0.0372 | 0.0017 | 0.0622 | −0.0362 | 0.0016 | 0.0610 | −0.0372 | 0.0017 | 0.0623 |
| | | $\delta$ | 0.3624 | 0.1346 | 0.2258 | 0.3790 | 0.1475 | 0.2470 | 0.3442 | 0.1211 | 0.2064 | 0.3692 | 0.1398 | 0.2338 | 0.3225 | 0.1062 | 0.1864 |
| | | $a$ | 0.1031 | 0.0180 | 0.3187 | 0.1156 | 0.0213 | 0.3285 | 0.0906 | 0.0150 | 0.3132 | 0.1050 | 0.0184 | 0.3206 | 0.0932 | 0.0157 | 0.3156 |
| | 40 | $b$ | −0.0780 | 0.0089 | 0.2048 | −0.0696 | 0.0077 | 0.2064 | −0.0863 | 0.0103 | 0.2051 | −0.0763 | 0.0086 | 0.2049 | −0.0866 | 0.0103 | 0.2065 |
| | | $\delta$ | 0.7317 | 0.5927 | 0.9007 | 0.8358 | 0.7821 | 1.0799 | 0.6180 | 0.4171 | 0.7011 | 0.7505 | 0.6241 | 0.9342 | 0.6282 | 0.4349 | 0.7433 |
| | | $a$ | 0.0423 | 0.0053 | 0.2295 | 0.0464 | 0.0058 | 0.2322 | 0.0381 | 0.0048 | 0.2286 | 0.0429 | 0.0053 | 0.2303 | 0.0389 | 0.0049 | 0.2297 |
| 1.2 | 80 | $b$ | −0.0347 | 0.0026 | 0.1457 | −0.0319 | 0.0024 | 0.1454 | −0.0376 | 0.0028 | 0.1452 | −0.0341 | 0.0026 | 0.1456 | −0.0376 | 0.0028 | 0.1455 |
| | | $\delta$ | 0.3935 | 0.1676 | 0.4409 | 0.4283 | 0.1995 | 0.4871 | 0.3573 | 0.1376 | 0.3851 | 0.4009 | 0.1741 | 0.4527 | 0.3549 | 0.1362 | 0.3910 |
| | | $a$ | 0.0339 | 0.0032 | 0.1765 | 0.0361 | 0.0033 | 0.1796 | 0.0317 | 0.0030 | 0.1745 | 0.0342 | 0.0032 | 0.1768 | 0.0321 | 0.0030 | 0.1751 |
| | 160 | $b$ | −0.0333 | 0.0021 | 0.1238 | −0.0316 | 0.0020 | 0.1235 | −0.0350 | 0.0023 | 0.1249 | −0.0330 | 0.0021 | 0.1237 | −0.0350 | 0.0023 | 0.1250 |
| | | $\delta$ | 0.3408 | 0.1224 | 0.3073 | 0.3661 | 0.1417 | 0.3402 | 0.3147 | 0.1039 | 0.2688 | 0.3464 | 0.1265 | 0.3149 | 0.3121 | 0.1023 | 0.2702 |
| | | $a$ | 0.0365 | 0.0058 | 0.2445 | 0.0440 | 0.0065 | 0.2480 | 0.0291 | 0.0052 | 0.2413 | 0.0377 | 0.0059 | 0.2444 | 0.0305 | 0.0053 | 0.2415 |
| | 40 | $b$ | −0.0165 | 0.0067 | 0.2990 | −0.0058 | 0.0067 | 0.3036 | −0.0271 | 0.0069 | 0.2941 | −0.0144 | 0.0066 | 0.2979 | −0.0271 | 0.0070 | 0.2965 |
| | | $\delta$ | 0.4806 | 0.3234 | 1.1672 | 0.5638 | 0.4389 | 1.3260 | 0.3958 | 0.2241 | 1.0117 | 0.4887 | 0.3334 | 1.1839 | 0.4399 | 0.2746 | 1.0958 |
| | | $a$ | 0.0158 | 0.0025 | 0.1799 | 0.0188 | 0.0027 | 0.1825 | 0.0128 | 0.0024 | 0.1786 | 0.0163 | 0.0026 | 0.1801 | 0.0133 | 0.0024 | 0.1791 |
| 3 | 80 | $b$ | −0.0097 | 0.0059 | 0.2519 | −0.0053 | 0.0053 | 0.2507 | −0.0140 | 0.0065 | 0.2519 | −0.0087 | 0.0056 | 0.2512 | −0.0142 | 0.0073 | 0.2523 |
| | | $\delta$ | 0.2242 | 0.0715 | 0.5563 | 0.2439 | 0.0842 | 0.5915 | 0.2043 | 0.0598 | 0.5135 | 0.2262 | 0.0727 | 0.5598 | 0.2141 | 0.0655 | 0.5382 |
| | | $a$ | 0.0110 | 0.0012 | 0.1251 | 0.0124 | 0.0012 | 0.1255 | 0.0096 | 0.0012 | 0.1245 | 0.0112 | 0.0012 | 0.1251 | 0.0098 | 0.0012 | 0.1247 |
| | 160 | $b$ | −0.0107 | 0.0019 | 0.1672 | −0.0089 | 0.0018 | 0.1683 | −0.0124 | 0.0019 | 0.1684 | −0.0103 | 0.0019 | 0.1673 | −0.0124 | 0.0019 | 0.1686 |
| | | $\delta$ | 0.1970 | 0.0475 | 0.3568 | 0.2092 | 0.0537 | 0.3804 | 0.1847 | 0.0417 | 0.3325 | 0.1982 | 0.0481 | 0.3603 | 0.1906 | 0.0445 | 0.3454 |

*5.1. Simulation Results*

Tables 3–6 show the results of the suggested techniques for calculating the point and interval parameter estimates. They offer the findings as well as some intriguing data. The following observations are permissible:

- The estimates are asymptotically unbiased since they are more accurate as the sample size increases.
- The parameter estimates come from the best unbiased estimator when the MSE value is near zero.
- As the sample size grows, the MSE declines for each estimate, demonstrating consistency between the various estimates.
- When the true value of $\delta$ increases, the bias, MSE, and length of the credible confidence interval (LCCI) of all estimates decrease.
- The MSE and LCCI for the Bayesian estimates with positive weight for the asymmetric loss function are smaller than the Bayesian estimates with negative weight for asymmetric loss function.
- The LCCI for estimates obtains its largest value, based on the suggested method, as the true values of the parameters increase.
- An entropy loss function with positive weight is better than the other loss functions.

*5.2. Representation Results*

Figures 4–7 show heatmap descriptions for the MSE results, where the bold color represents the highest values of MSE and the white color represents the lowest values of MSE.

The X-label belongs to SELFj, ($j$ = 1, 2, 3) which are the MSE of Bayes estimates based on SELF with different parameters;

LINEXaj, ($j$ =1, 2, 3) are the MSE of Bayes estimates based on LINEX ($c = -1.5$) with different parameters;

LINEXbj, ($j$ =1, 2, 3) are the MSE of Bayes estimates based on LINEX ($c = 1.5$) with different parameters;

ELFaj, ($j$ =1, 2, 3) are the MSE of Bayes estimates based on ELF ($c = -1.5$) with different parameters;

ELFbj, ($j$ =1, 2, 3) are the MSE of Bayes estimates based on ELF ($c = 1.5$) with different parameters.

The Y-label belongs to cases and sample sizes, where C1n1 for $\delta = 0.5$ and $n = 40$; C1n2 for $\delta = 0.5$ and $n = 80$; C1n3 for $\delta = 0.5$ and $n = 80$.

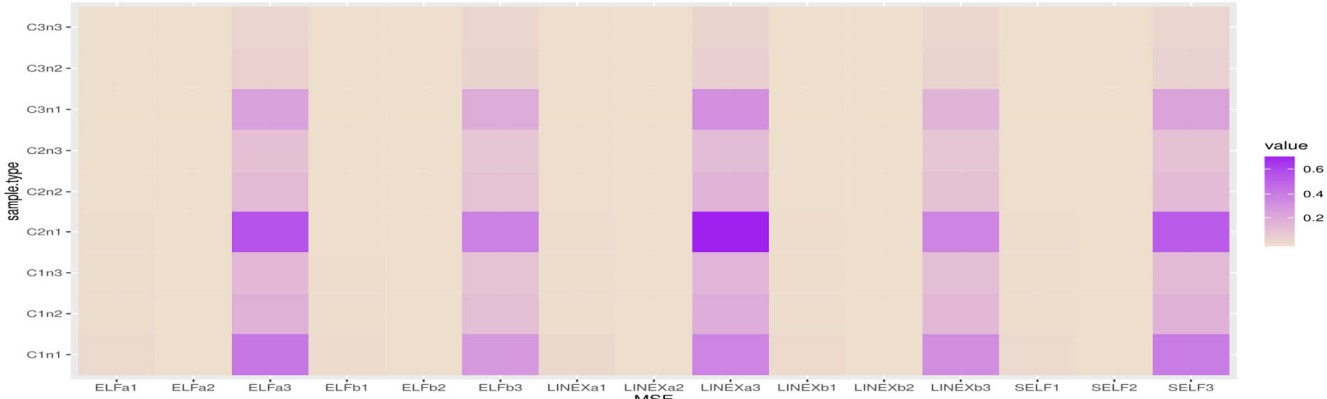

**Figure 4.** Heatmap for MSE when $a = 0.5$, $b = 0.6$.

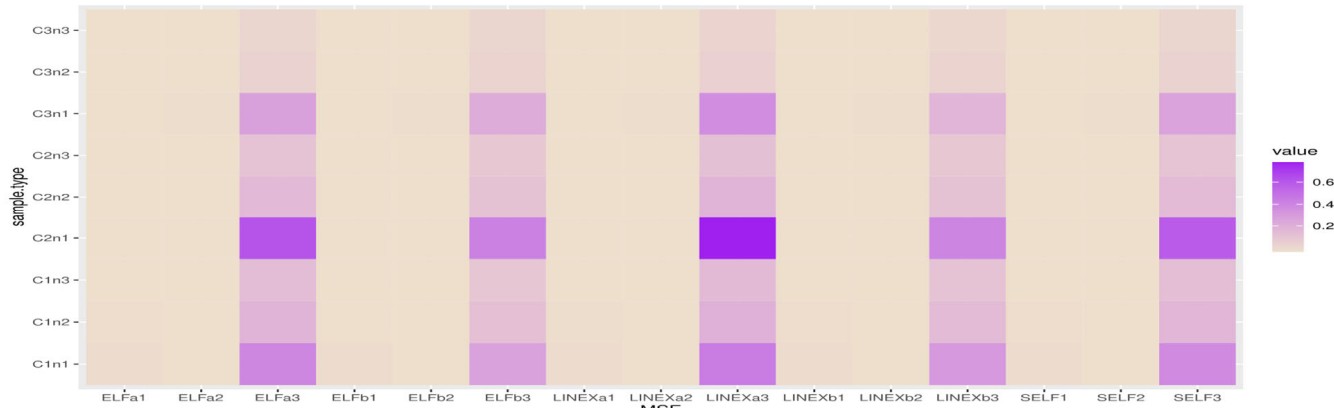

**Figure 5.** Heatmap for MSE when $a = 0.5$, $b = 1.7$.

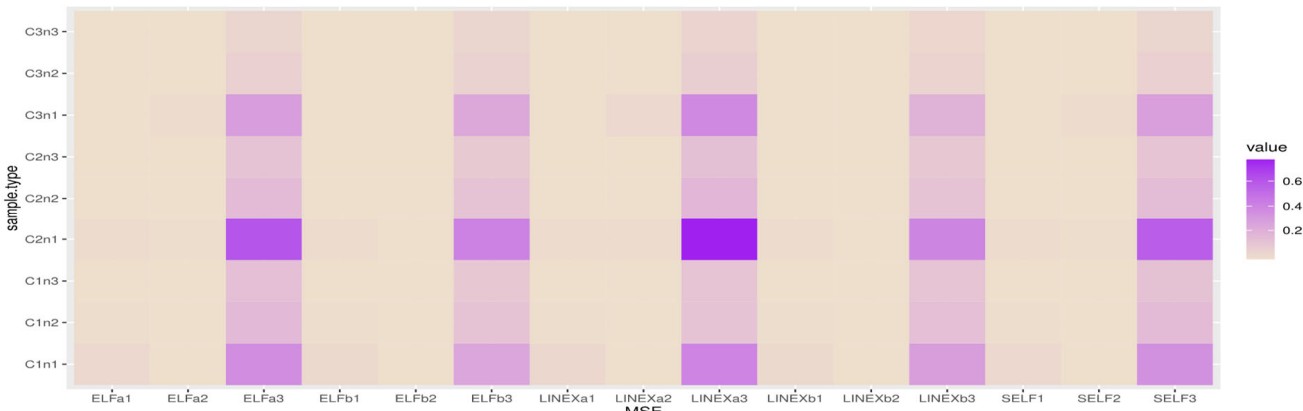

**Figure 6.** Heatmap for MSE when $a = 2$, $b = 0.6$.

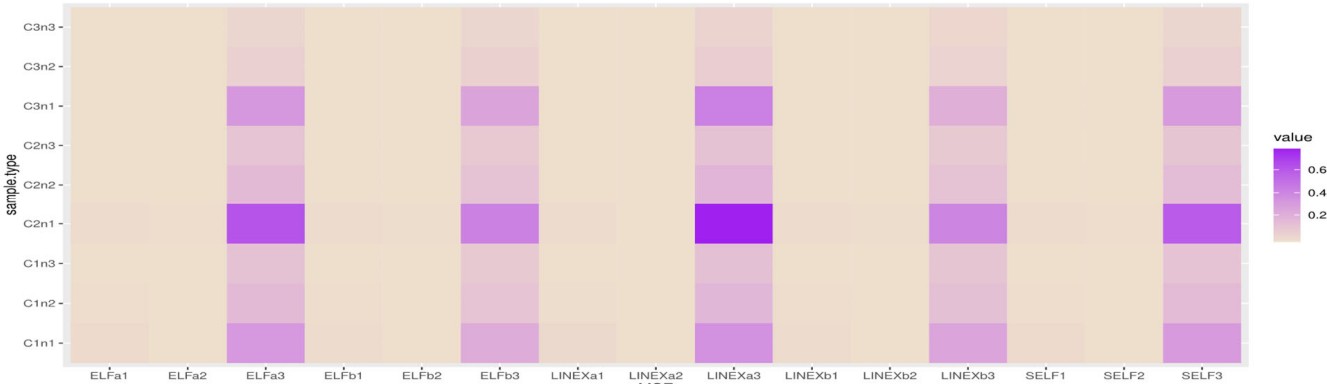

**Figure 7.** Heatmap for MSE when $a = 2$, $b = 1.7$.

## 6. Application of Real Data

This section analyses two real-world datasets to show the adaptability and practical application of the UPBXD. The UPBXD is compared with the following models: unit-exponentiated half-logistic (UEHL) [23], Type II power Topp–Leone exponential (TIIPTLE) [34], Topp–Leone generalized exponential (TLGE) [35], Kumaraswamy (K), Beta, unit Weibull (UW), and Marshall–Olkin–Kumaraswamy (MOK). Two actual COVID-19 mortality rate datasets from Saudi Arabia and the United Kingdom are provided in this section to evaluate the UPBXD goodness of fit. The two real datasets were utilized to estimate the unknown parameters of the specified models using the maximum likelihood and Bayesian approaches. Kolmogorov–Smirnov statistics (KSS) with *p*-value, Cramer–von

Mises statistics (WS), and Anderson–Darling statistics (AS) were used to compare all of the models.

### 6.1. Analysis for First Data

Data set I: The first set of data shows Saudi Arabia's COVID-19 mortality rates over a 36-day period (22 July 2021 to 26 August 2021). The information is as follows: 0.1310, 0.1319, 0.1497, 0.1504, 0.1686, 0.1689, 0.1706, 0.1716, 0.1879, 0.1890, 0.1924, 0.1951, 0.2063, 0.2077, 0.2091, 0.2113, 0.2126, 0.2140, 0.2167, 0.2249, 0.2259, 0.2271, 0.2278, 0.2314, 0.2329, 0.2347, 0.2353, 0.2375, 0.2452, 0.2487, 0.2666, 0.2674, 0.2683, 0.2711, 0.2752, 0.2962. Table 7 shows the ML estimate of parameters with their standard errors (SEs) for each distribution and obtained the goodness of fit measures as KSS, WS, and AD. By the results shown in Table 7, we are able to see that the UPBXD is better than the other distributions, such as TLPTLE, TLGE, K, Beta, UW, UEHL, and MOK, for COVID-19 mortality rates in the Saudi Arabia data set.

**Table 7.** ML estimates with SE and goodness of fit statistics: Saudi Arabia data set.

|  |  | $a$ | $b$ | $\delta$ | KSS | $p$-Value | WS | AS |
|---|---|---|---|---|---|---|---|---|
| UPBXD | Estimates | 2.2717 | 0.5609 | 2813.2886 | 0.0778 | 0.9693 | 0.0327 | 0.2364 |
|  | SE | 0.0708 | 0.0692 | 469.2354 |  |  |  |  |
| TLPTLE | Estimates | 693.1774 | 0.6471 | 0.6476 | 0.0938 | 0.8800 | 0.0479 | 0.3110 |
|  | SE | 1626.8445 | 0.0939 | 0.8346 |  |  |  |  |
| TLGE | Estimates | 0.3682 | 20.4075 | 179.7044 | 0.1403 | 0.4378 | 0.0965 | 0.5911 |
|  | SE | 0.2843 | 2.7038 | 178.4010 |  |  |  |  |
| K | Estimates | 3.3085 | 125.2161 |  | 0.1821 | 0.1621 | 0.0421 | 0.2793 |
|  | SE | 0.2821 | 49.4480 |  |  |  |  |  |
| Beta | Estimates | 20.8174 | 76.5218 |  | 0.1127 | 0.7089 | 0.0636 | 0.3992 |
|  | SE | 4.8690 | 18.0555 |  |  |  |  |  |
| UW | Estimates | 0.0203 | 7.7557 |  | 0.1633 | 0.2624 | 0.1824 | 1.0950 |
|  | SE | 0.0110 | 0.9132 |  |  |  |  |  |
| UEHL | Estimates | 6.0655 | 3670.3422 |  | 0.0792 | 0.9641 | 0.0330 | 0.2393 |
|  | SE | 0.7918 | 405.8862 |  |  |  |  |  |
| MOK | Estimates | 703.3130 | 1.3097 | 45.4476 | 0.0811 | 0.9567 | 0.0336 | 0.2554 |
|  | SE | 4615.4897 | 1.5712 | 72.4483 |  |  |  |  |

As can be seen, the TLPTLE, TLGE, K, Beta, UW, UEHL, and MOK distributions work well for modelling the COVID-19 mortality rates indicated in the Saudi Arabia data set, but that the UPBXD is the best. This is based on a significance level of 0.05. Figure 8 illustrates the estimated CDF in the red line with empirical CDF in the black line. It also shows the probability–probability (PP) plots of the UPBXD in the red line, also known as "parametric plots", for the COVID-19 mortality rates of the Saudi Arabia data set, which demonstrate the empirical findings, reported in Table 7 and the empirical CDF line the (black) with the estimated CDF line (red).

Figure 9 shows three plots of COVID-19 mortality rates for the Saudi Arabia data set, where the left is a boxplot of data that explains that the data have no outlier values, the center is a TTT plot of data that explains this data set is increasing, and the right is a hazard estimated plot line that indicates the HF is increasing.

### 6.2. Analysis for Second Data

Data set II: The second set of data shows the United Kingdom COVID-19 mortality rates over a 28-day period (1 January 2022 to 28 January 2022). The information is as follows: 0.1484, 0.1174, 0.0522, 0.0296, 0.0339, 0.2274, 0.1555, 0.1530, 0.2079, 0.0640, 0.1407, 0.2463, 0.2569, 0.2150, 0.1723, 0.1823, 0.1807, 0.1823, 0.2736, 0.2228, 0.2036, 0.1767, 0.1814, 0.1361, 0.1620, 0.2639, 0.2067, 0.2008.

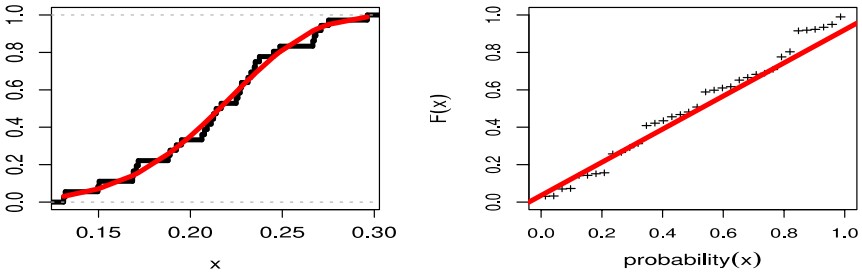

**Figure 8.** The CDF plot with empirical line and PP plot for Saudi Arabia data set.

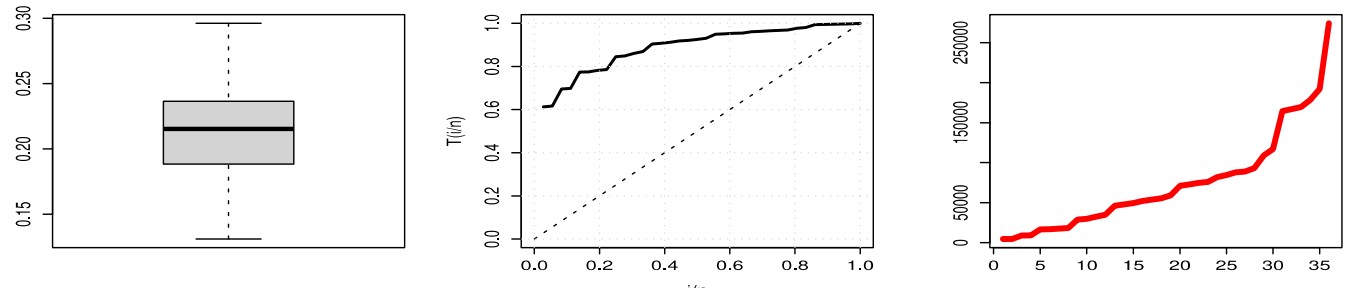

**Figure 9.** Boxplot, TTT plots and hazard line of UPBXD plot for Saudi Arabia data set.

Table 8 shows the ML estimate of parameters for each distribution and obtained the goodness of fit measures as KSS, WS, and AD. By the results of Table 8, we can see that the UPBXD is better than the other distributions, such as TLPTLE, TLGE, K, Beta, UW, UEHL, and MOK, for COVID-19 mortality rates in the United Kingdom data set. Additionally, we can see that the TLPTLE, TLGE, K, Beta, UW, UEHL, and MOK distributions work well for modelling the COVID-19 mortality rates of the United Kingdom data set, though the UPBXD is the best. This is based on a significance level of 0.05.

**Table 8.** Estimates with SE and goodness of fit statistics of ML: The United Kingdom data set.

| | | *a* | *b* | *δ* | **KSS** | *p*-**Value** | **WS** | **AS** |
|---|---|---|---|---|---|---|---|---|
| UPBXD | Estimates | 2.3471 | 0.3354 | 2154.8742 | 0.1100 | 0.8512 | 0.1115 | 0.7238 |
| | SE | 0.0716 | 0.0478 | 402.9847 | | | | |
| TLPTLE | Estimates | 3760.8372 | 0.1127 | 0.0216 | 0.2571 | 0.0404 | 0.4656 | 2.6059 |
| | SE | 5117.2571 | 0.0287 | 0.0151 | | | | |
| TLGE | Estimates | 0.2968 | 10.3896 | 13.8963 | 0.2097 | 0.1471 | 0.2804 | 1.6718 |
| | SE | 0.3076 | 2.3410 | 15.4726 | | | | |
| K | Estimates | 2.9163 | 125.0007 | | 0.1329 | 0.6570 | 0.1492 | 0.9456 |
| | SE | 0.4689 | 94.0394 | | | | | |
| Beta | Estimates | 3.9277 | 19.1899 | | 0.1925 | 0.2202 | 0.2609 | 1.5673 |
| | SE | 1.0090 | 5.1885 | | | | | |

**Table 8.** *Cont.*

|      |           | *a*    | *b*      | *δ*     | KSS    | *p*-Value | WS     | AS     |
|------|-----------|--------|----------|---------|--------|-----------|--------|--------|
| UW   | Estimates | 0.0904 | 3.2548   |         | 0.2468 | 0.0548    | 0.4622 | 2.5905 |
|      | SE        | 0.0386 | 0.4231   |         |        |           |        |        |
| UEHL | Estimates | 2.9789 | 69.5723  |         | 0.1294 | 0.6888    | 0.1476 | 0.9369 |
|      | SE        | 0.4808 | 53.4192  |         |        |           |        |        |
| MOK  | Estimates | 0.0124 | 3.5483   | 6.7014  | 0.1506 | 0.5020    | 0.3117 | 1.8656 |
|      | SE        | 0.0299 | 0.6026   | 14.9920 |        |           |        |        |

Figure 10 illustrates the PP plots for the COVID-19 mortality rates of the United Kingdom data set, which demonstrate the empirical findings reported in Table 8 and the empirical CDF line (black) with the estimated CDF line (red). Figure 11 shows three plots of COVID-19 mortality rates for the United Kingdom data set, where the left is a boxplot of data that explains that these data have no outlier values, the center is a TTT plot of data that explains that these data are increasing, and the right is a hazard estimated plot line that indicates the hazard is increasing.

*6.3. Data Analysis via Bayesian Method*

Here, we analyze data sets presented in previous sub-sections using the proposed Bayesian estimation method.

The Bayesian estimation parameters of UPBXD for each of the data sets, respectively are given in Table 9. The Bayesian estimates of UPBXD parameters under SELF and the corresponding SEs are calculated. The lower and upper HPD intervals are also calculated.

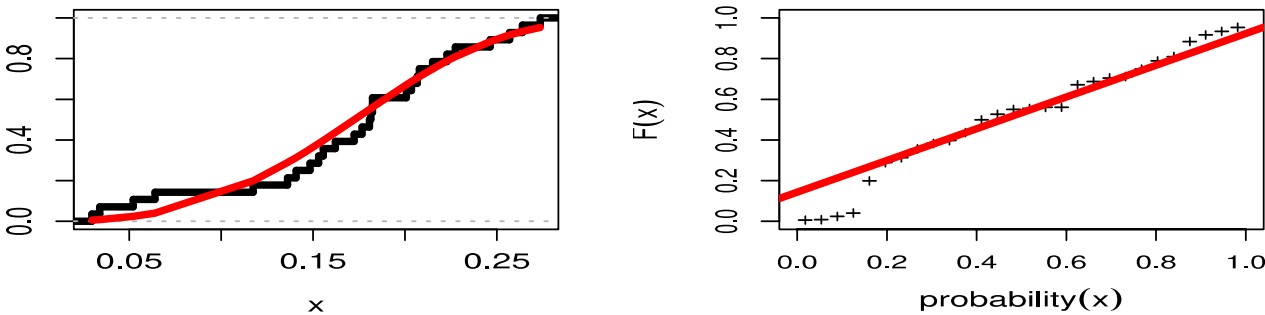

**Figure 10.** The CDF plot with empirical line and PP plot for the United Kingdom data set.

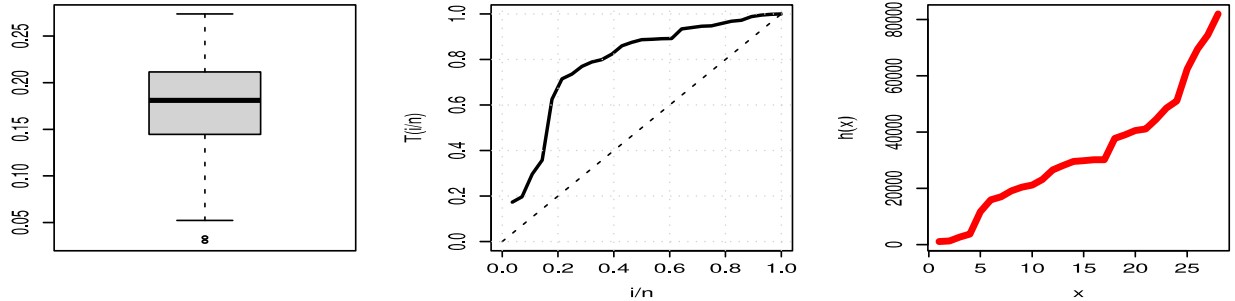

**Figure 11.** Boxplot, TTT plots and hazard line of UPBXD plot for the United Kingdom data set.

**Table 9.** Bayesian estimation based on SELF for parameters of UPBXD.

| Data | | Estimates | SE | Lower | Upper |
|---|---|---|---|---|---|
| Saudi Arabia | $a$ | 2.3235 | 0.0619 | 1.9541 | 2.7490 |
| | $b$ | 0.5556 | 0.0558 | 0.3990 | 0.7320 |
| | $\delta$ | 2979.2033 | 2.4931 | 2974.2166 | 2984.1337 |
| The United Kingdom | $a$ | 2.3852 | 0.0560 | 2.0108 | 2.8278 |
| | $b$ | 0.3353 | 0.0314 | 0.2064 | 0.4830 |
| | $\delta$ | 2154.8743 | 0.0787 | 2154.7169 | 2155.0299 |

Figures 12 and 13 display the trace plot of the UPBXD's parameter values for the MCMC finding.

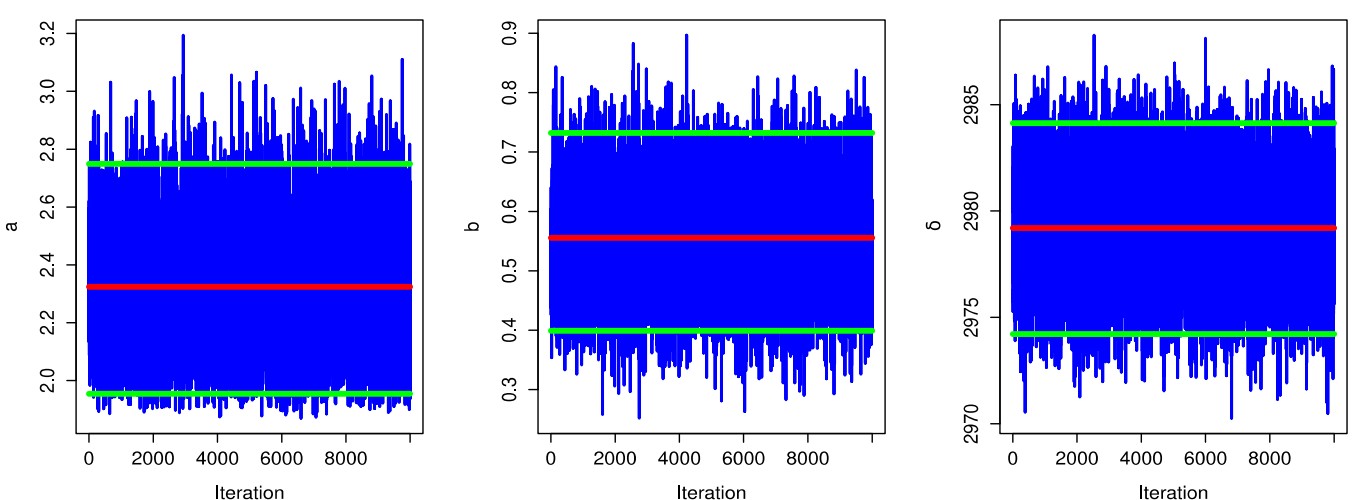

**Figure 12.** Trace plots of MCMC results with interval limit line for Saudi Arabia data set.

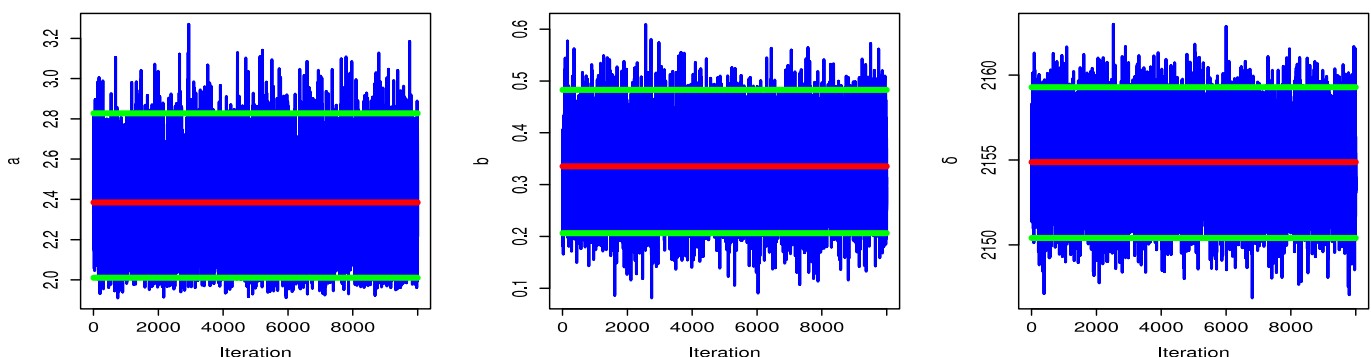

**Figure 13.** Trace plots of MCMC results with interval limit line for the United Kingdom data set.

The autocorrelation function (ACF) is generated as shown in Figures 14 and 15. Figures 16 and 17 demonstrate the symmetric normal distribution of the posterior density for the parameters of the UPBXD.

Figures 18 and 19 display the parameter convergence charts for UPBXD draws as well as the parameter random draw plot, respectively.

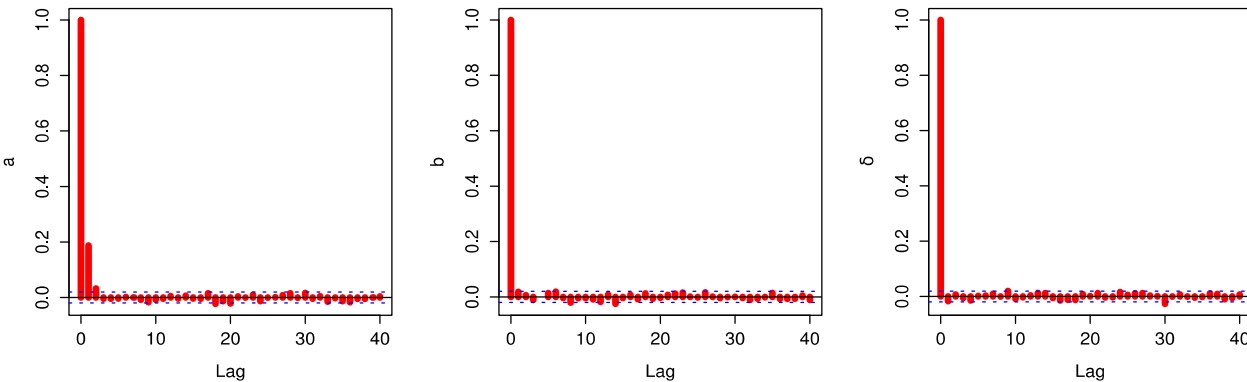

**Figure 14.** The ACF plot of MCMC results for Saudi Arabia data set.

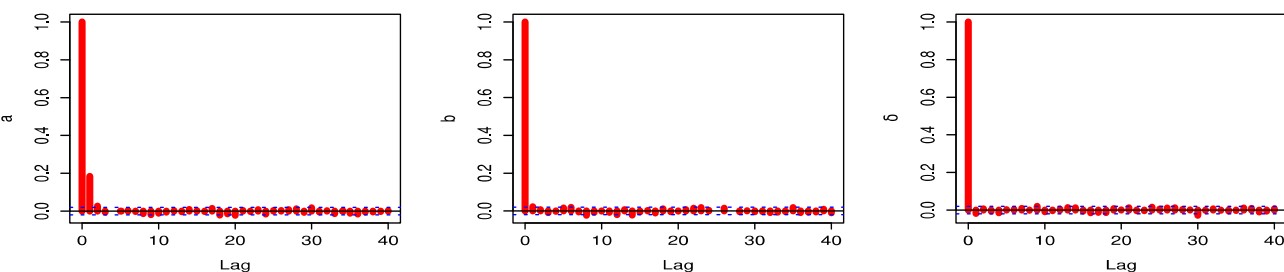

**Figure 15.** ACF plot of MCMC results for the United Kingdom data set.

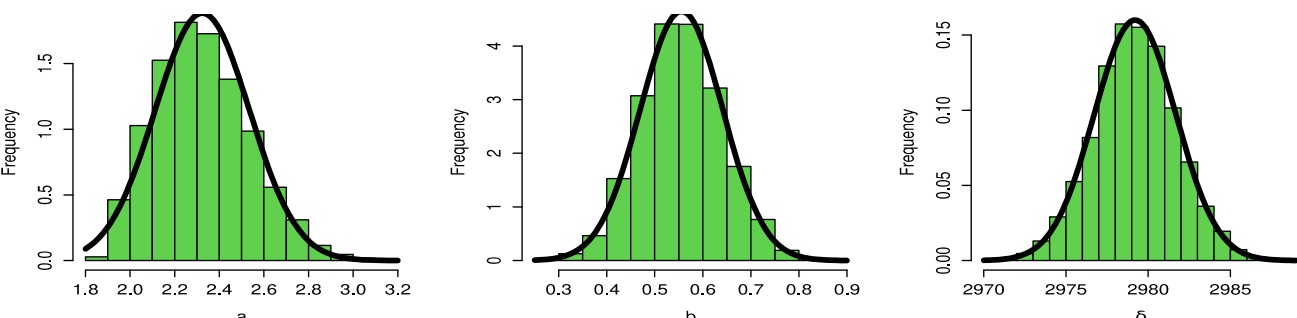

**Figure 16.** Histogram plots of MCMC results for Saudi Arabia data set.

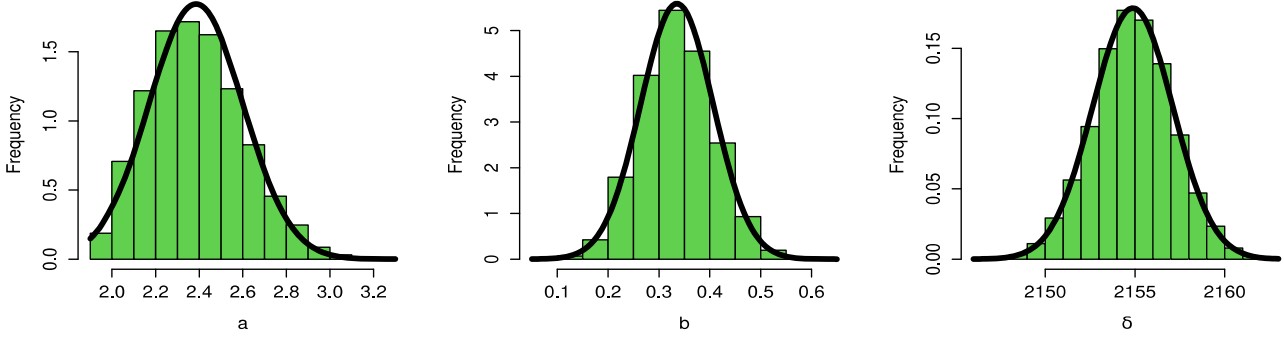

**Figure 17.** Histogram plots of MCMC results for the United Kingdom data set.

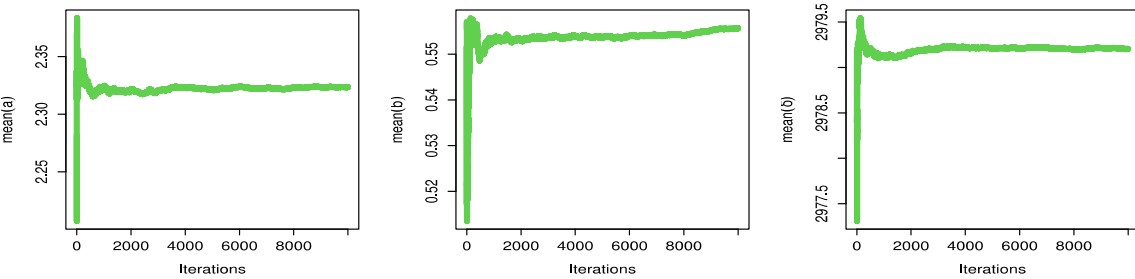

**Figure 18.** Convergence lines of MCMC results for Saudi Arabia data set.

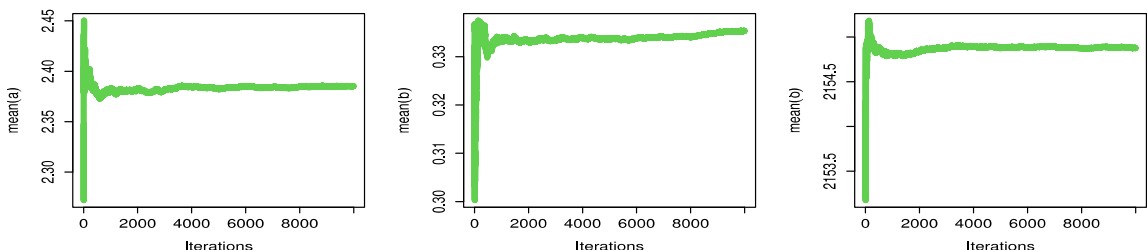

**Figure 19.** Convergence lines of MCMC results for the United Kingdom data set.

## 7. Conclusions

This article focuses on a three-parameter unit distribution created based on the power Burr X distribution and called the UPBXD. The statistical properties of the UPBXD have been derived and expressed in closed forms. The presented unit distribution can be used as a statistical tool to model different types of HFs, including those that are bathtub, increasing and unimodally shaped. Its important features are carefully studied, including the analytical expression of moments, quantile function, incomplete moments, stochastic ordering, and stress–strength reliability. Moreover, the uncertainty-measuring metrics Rényi, Havrda, and Charvat as well as d-generalized entropy were obtained. The UPBXD parameters have been estimated utilizing ML approach as well as Bayesian estimation approach with different loss functions. Additionally, Bayesian credible intervals were constructed based on the marginal posterior distribution. For some difficult calculations, the Markov chain Monte Carlo method was used. To assess how various estimates work, simulation studies based on various sample sizes have been carried out. In light of the simulation study's findings, it was found that the Bayesian-based symmetric loss function and LINEX loss function techniques work quite effectively for estimating the UPBXD parameters. Bayesian estimates under an entropy loss function with positive weight are superior to those under other loss functions. The MSE and length of the credible confidence interval for Bayesian estimates with positive weight are smaller than the corresponding values with negative weight. Finally, two actual COVID-19 mortality rate data sets from Saudi Arabia and the United Kingdom have been analyzed and discussed to illustrate the notability of the UPBXD. The UPBXD gives superior fits over several other competing models, as shown by a real data application. Future discussions can be expanded on the use of Bayesian estimation in stress–strength reliability for the UPBXD based on some sampling techniques [36–38]. Furthermore, the proposed methodology can be expanded in multivariate and Bivariate case as [39–41].

**Author Contributions:** Conceptualization, A.S.H. and E.M.A.; Methodology, A.F., A.S.H., H.B. and E.M.A.; software, A.S.H. and E.M.A.; validation, A.S.H. and H.B.; formal analysis, A.F. and H.B.; investigation, A.F. and H.B.; data curation, A.S.H. and E.M.A.; writing—original draft, A.F., A.S.H., H.B. and E.M.A.; writing—review and editing, A.S.H., H.B. and E.M.A. All authors have read and agreed to the published version of the manuscript.

**Funding:** This research received no external funding.

**Data Availability Statement:** The data sets have been provided in Section 6.

**Acknowledgments:** The authors would like to thank the editor and the anonymous referees for their efforts, wise observations, and constructive criticism, which greatly enhanced the manuscript's contents.

**Conflicts of Interest:** The authors declare no conflict of interest.

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
