# Peer review of "Bayesian Inference and Data Analysis of the Unit–Power Burr X Distribution"

_axioms, doi:10.3390/axioms12030297_

Round 1

Reviewer 1 Report

The paper proposes and studies a new family of distributions: the unit-power Burr X distribution (UPBXD). The paper is well written. However, the manuscript should be revised according to the following comments:

1. The presentation of the plot in Figure 1 should be improved (size of letters on axes, legends, and the thickness of the lines of the PDFs).

2. Explain in more detail the role (interpretation) that each parameter plays in the new distribution.

3. Line 129: a,b,\delta \in ... R^+ ?

4. Lines 260 and 262: put transpose symbol to the vector (a,b,\delta).

5. In general, the names of the axes in the figures all over the paper are in very small font sizes. Please improve this.

6. Conclusion Section should be expanded by adding comment on the research gaps, research objectives, methodology and future direction. For instance: can your methodology be extended to the multivariate case?

Author Response

Response to Reviewer 1 Comments

Manuscript entitled

“Bayesian Inference and Data Analysis of the Unit-Power Burr X Distribution”

First of all, we would like to express our sincere thanks and appreciation to the reviewers for their comments, which improved the paper. All the corrected parts are written with red color within the main text.

Comments

  1. The presentation of the plot in Figure 1 should be improved (size of letters on axes, legends, and the thickness of the lines of the PDFs).

Response 1:
Done, Figure 1 has been improved.

=======================

  1. Explain in more detail the role (interpretation) that each parameter plays in the new distribution

 Response 2:
After checking, we note that the  parameter is responsible for the bathtub and reversed J-shaped shape given that the other two parameters ( and ) are less than one. The  parameter is responsible for the J-shaped shape where  and  .

=======================

  1. Line 129: a,b,\delta \in ... R^+ ?

Response 3:
Ok, we replace it with greater than (>0)

=======================

  1. Lines 260 and 262: put transpose symbol to the vector (a,b,\delta).

Response 4:
Ok, thanks we done 

=======================

  1. 5. In general, the names of the axes in the figures all over the paper are in very small font sizes. Please improve this.

Response 5:
All figures have been redrawn and reorganized

=======================

  1. Conclusion Section should be expanded by adding comment on the research gaps, research objectives, methodology and future direction. For instance: can your methodology be extended to the multivariate case?

Response 6:
Done

=======================

All corrected parts are done with red color and green highlighted color

Best Regards

Reviewer 2 Report

Please find my comments in the attached pdf.

Author Response

Response to Reviewer 2 Comments

Manuscript entitled

“Bayesian Inference and Data Analysis of the Unit-Power Burr X Distribution”

First of all, we would like to express our sincere thanks and appreciation to the reviewers for their comments, which improved the paper. All the corrected parts are written with red color within the main text.

Comments

  1. In line 45 (and in a few instances later), does the authors intend to write a, b ∈ R+? It is just showing a rectangle instead of R

Response 1:
Ok, we thanks, we write > 0, than R+  

=======================

  1. In line 129-130, should it be F(w) = 1, for w ≥ 1 instead of w ≤ 1?

Response 2:
Ok thanks, we correct it  

=======================

  1. For figure 1, few more illustrations of the shape of the pdf and hazard function may be helpful for readers. In fact, if instead of showing several pdfs (and hazard function) in one single plot the authors may want to show them in subplots, varying only one parameter in each subplot. This will provide a better insight on the effect of each individual parameters a, b and δ.

Response 3:
After checking, we note that the  parameter is responsible for the bathtub and reversed J-shaped shape given that the other two parameters (  and ) are less than one. The  parameter is responsible for the J-shaped shape where  and   . A figures has been made but there are no new general shapes of this figures. The plots of the pdf and hazard function are redrawn with subplots in figure 1 and 2

=======================

  1. Similar to the previous comment, table 1 perhaps should be accompanied with a plot the shows the moments as a function of the three parameters. In fact, the authors can consider obtaining and showing the moments of the distribution for the exact parameters that are used in figure 1. Same applies for table 2

Response 4:
Ok, values in table 1 and 2 have been reconsidered with the selected values used in figures 1with red color and green highlighted color

=======================

  1. In line 188, is there an extra comma in the upper incomplete gamma function γ(・, t)?

Response 5:

Ok, thanks, extra comma are deleted

=======================

  1. Line 229, 236 - the authors define the stochastic ordering in terms of f1(w)/f2(w) as decreasing function of ω instead of w. Is ω same as w or is it just a type-setting error? Please provide appropriate citation for this definition of stochastic ordering. Is there any bound on f1 and f2?

Response 6:

Ok, thanks, it just misprints error, and we add cited reference. f1 and f2 are independent

=======================

  1. In line 281, ”posterior distribution of ...”, missing parameter

Response 7:

Ok, thanks we write the missing parameter

=======================

  1. In line 284, what is theta?

Response 8:

Thanks, it just mistake, it the symbol   so we correct it

=======================

  1. Figure 2-5 need few modifications. The aspect ratio of the plots are skewed which should be corrected (this issue of aspect ratio needs to addressed for all figures in the manuscript). Some description of the axis tick labels (MSE11, MSE12, C1n1 etc., they are defined later) should be put in figure caption. The legend also needs a proper description.

Response 9:

We have tried to improve the overall quality and look for these Figures as possible. Also, the description of the axis tick labels (MSE11, MSE12, C1n1 etc. ) are all defined before the figures (as seen in sub-section 5.2)

 =======================

  1. Table 3-6 can be moved to appendix and should be accompanied with a visual description

of the numbers using figures.

Response 10:

The tables are moved to the appendix, and figures 3,4,5,6 provide description of results in table3 3-6

 =======================

  1. Line 438-439, MSE3j has been defined twice, once for Linex 2 and then for ELF 1. Perhaps the authors intend MSE4j for ELF 1.

Response 11:

Ok, thanks for this observation, we correct it

=======================

  1. In figure 6, the caption should be elaborated with more information such as the red line represents the fitted distribution and the black step function is the empirical CDF. The plot should have a legend. Figure 7 caption needs more description. TTT plot is never explained. Same comment applies to figure 12,13.

Response 12:

Ok, thanks we done

 =======================

  1. According to table 7, UEHL has the lowest WS value, but the WS value corresponding to UPBX is highlighted in bold. Is that intentional or a typo?

Response 12:

Ok, a thousand thanks for this good note

 it's a typo. The Tables of results has been reviewed well.  

=======================

  1. Line 543, is it a two-parameter or three parameter distribution? (a, b, δ)

Response 12:

Ok, thanks for this observation it's just a typo, the distribution with three parameter and we correct it

=======================

  1. Line 546: “It was found that the UPBX model is proper for modeling right-skewed data sets of a bathtub shape” - an illustration (simulation or real case study) needs to be presented before making such claim.

Response 12:

Ok thanks, we arrange this statement

 =======================

  1. feel that the real case studies does not provide a sufficient empirical justification as to the benefit of UPBX distribution over others. In both the cases UEHL performs very similarly to UPBX with one less parameter to estimate, which can potentially save a significant amount of computation. The authors do not include any discussion on computation time and/or budget in regards to the estimation of parameter and the robustness of the ML estimates (robustness of the optimization). The studies only illustrate that the UPBX is another distribution in the bag of distributions for some problems without showing example where it can be beneficial the most. It is recommended to find another use-case to show UPBX’s real advantage over others, otherwise

the claims on the usefulness of UPBX should be modified accordingly

Response 12:

Computation time: Mathematical study takes two weeks; Computational numerical study takes 10- 14 days; Computational revision study takes 7 days.

budget in regards to the estimation of parameter and the robustness of the ML estimates (robustness of the optimization) is Laptop DELL with processor I9 in 13th.

Finally, two actual COVID-19 mortality rate data from Saudi Arabia and the United Kingdom have been analyzed and discussed to illustrate the notability of the UPBXD. The UPBXD gives superior fits than several other competing models, as shown by a real data application.

=======================

All corrected parts are done with red color and green highlighted color

Best Regards

Round 2

Reviewer 1 Report

The paper has been considerable improved.

Author Response

Thanks a lot 

Reviewer 2 Report

This manuscript has improved after the first revision. Further modifications are suggested below:

- Please present the numbers in Table 1 and 2 in suitable graphs.

- In Figure 1 and 2, please add more PDFs and HFs by varying a and b keeping delta fixed in each subplot.

Author Response

Response to Reviewer 2 Comments (2nd round)

Manuscript entitled

“Bayesian Inference and Data Analysis of the Unit-Power Burr X Distribution”

First of all, we would like to express our sincere thanks and appreciation to the reviewers for their comments, which improved the paper. All the corrected parts are written with red color within the main text.

Comments and Suggestions for Authors

This manuscript has improved after the first revision. Further modifications are suggested below:

  1. Please present the numbers in Table 1 and 2 in suitable graphs.

Response 1:
Thank you very much for this comment, we done   

=======================

  1. In Figure 1 and 2, please add more PDFs and HFs by varying a and b keeping delta fixed in each subplot.

Response 2:
Thank you very much for this comment, we done   

=======================

All corrected parts are done with red color and green highlighted color

Best Regards
